

**Methane emissions from a sediment-deposited island in a Lancang-Mekong**
**reservoir**
Wenqing Shi[1,2], Qiuwen Chen[1,2], Jianyun Zhang[1,3], Cheng, Chen[2], Yuchen Chen[2],Yuyu
Ji[2,4], Juhua Yu[1,2], Bryce R. Van Dam[5]
[1]State Key Laboratory of Hydrology-Water Resources & Hydraulic Engineering,
Nanjing Hydraulic Research Institute, China
[2]Center for Eco-Environment Research, Nanjing Hydraulic Research Institute, China
[3]Research Center for Climate Change, Ministry of Water Resources, China
[4]College of Water Conservancy and Hydropower Engineering, Hohai University, China
[5]Institute of Marine Sciences, University of North Carolina at Chapel Hill, USA
Correspondence to: Qiuwen Chen (qwchen@nhri.cn)



**Abstract**
In dammed rivers, sediment accumulation creates potential methane emission hotspots,
which have been extensively studied in forebays. However, methane emissions from
sidebays remain poorly understood. We investigated methane emissions from a
sediment-deposited island situated in the sidebay of the Manwan Reservoir, Lancang-
Mekong River. High methane emissions (maximum 10.4 mg $h^{-1}m^{-2}$) were observed at
the island center, while a ring-like zone of low-to-negative methane emission was
discovered around the island edge, whose flux varied between -0.2–1.6 mg $h^{-1}m^{-2}$. The
ring-like zone accounted for 89.1 % of the island area, of which 9.1 % was a methane
sink zone. Microbial processes in the hyporheic zone, regulated by hydrological
variations, were responsible for the low methane flux in this area. Under reservoir
operation, frequent water level fluctuations enhanced hyporheic exchange and created
redox gradients along the hyporheic flow path. Dissolved oxygen in hyporheic water
decreased from 4.80 mg $L^{-1}$ at the island bank edge to 0.43 mg $L^{-1}$ at the center, which
in turn decreased methanogen abundance for methane production and increased
methanotroph abundance for methane oxidation at the ring-like zone. This study
quantified the methane emissions from sediment deposited islands in the reservoir and
helps to evaluate the global warming effects of hydropower systems.



## 1 Introduction

Natural rivers form continuous ecosystems, in which physical and chemical factors drive biological processes from headwaters to river deltas (Butman and Raymond, 2011; Wilkinson et al., 2015). Along this continuum, rivers receive terrestrial organic carbon (OC) and deliver it to the ocean at a global average rate of approximately 400–900 Tg OC per year (Butturini et al., 2016; Seitzinger et al., 2005; Ran et al., 2013). In the past two decades, many rivers have become intensively regulated by dams for a variety of purposes, including improved navigation, water supply, flood control, and hydropower production (Maavara et al., 2015). These engineering works decrease water velocity, converting rivers into a series of lentic reservoirs, where sediment accumulates in forebays and sidebay islands (Maeck et al., 2013). Globally, the sediment accumulation process has reduced the river-to-ocean flux of terrestrial OC by 26 % (Syvitski et al., 2005).

Settling particles aggregate to form cohesive sediment layers, which often become anoxic after oxygen is consumed but not replenished through diffusive exchange (Rubol et al., 2013; Maeck et al., 2013). Subsequently, large amounts of methane may be produced and released into the atmosphere (Thornton et al., 1990; Maeck et al., 2013; Wilkinson et al., 2015), thereby reducing the green credentials of hydropower. This issue has received considerable attention in dammed rivers (Giles, 2006; Hu and Cheng, 2013). Maeck et al. (2013) identified reservoirs as methane emission hotspots by comparing reservoir and riverine reaches, and estimated that global methane emissions have increased by 7 % due to sedimentation in dammed rivers. In sidebays, the





deposited sediments often form hyporheic zones, where water, heat, nutrients and
chemicals are exchanged and many biogeochemical reactions preferentially occur
(Tonina and Buffington, 2011; Cardenas and Markowski, 2010), potentially emitting
large amounts of greenhouse gases. Previous studies have mainly focused on methane
emissions from dam forebays (Yang et al., 2013; DelSontro et al., 2010; DelSontro et
al., 2011), while the understandings of methane emissions from sediments deposited in
sidebays remain poor.
In reservoirs, frequent water level fluctuations often occur following hydropower
production demands, which enhances hyporheic exchange by driving water flow in and
out of reservoir sidebays (Tonina and Buffington, 2011; Hucks Sawyer et al., 2009).
This may lead to changes in the redox conditions of sidebay sediments. Zarnetske found
a redox gradient along the hyporheic flow paths in a third-order stream in the
Willamette River basin, USA (Zarnetske et al., 2011a). Methane from sediments is
mainly produced by anaerobic methanogens, and is consumed by aerobic
methanotrophs (Borrel et al., 2011). We suppose the shift in sediment redox conditions
may affect the microbial processes, thereby altering the methane emission scheme.
In this study, methane emissions from a sediment-deposited island were investigated
in the sidebay of Manwan Reservoir, Lancang-Mekong River. Monitoring wells were
established to probe hyporheic exchange and redox gradients across the island.
Methanogen and methanotroph abundances in the sediment were analyzed using
quantitative polymerase chain reaction (qPCR) to reveal the associated molecular
mechanism. The objective of this study was to explore methane emissions from





sediment-deposited zones in a sidebay of a dammed river, with the goal to guide future
mitigation of the global warming effects of hydropower development.
**2    Methods**
**2.1 Study area**
The Lancang-Mekong River, a trans-boundary river in Southeast Asia, originates from
the Tibetan Plateau and discharges into the South China Sea. It has a length of 4909 km,
a watershed area of 760,000 km$^2$, and a mean annual runoff of 457 km$^3$ at a discharge
of 14,500 m$^3$ s$^{-1}$ (Li et al., 2013). The Lancang-Mekong Basin can be divided into two
parts: the "upper basin" in China, and the "lower basin" from Yunnan in China to the
Southeast Asia. Until now, seven dams have been built for hydropower production in
the upper Lancang-Mekong River in China, including Miaowei, Gongguoqiao,
Xiaowan, Manwan, Dachaoshan, Nuozhadu and Jinghong. The locations and main
features of these dams were shown in Fig. 1 and Table S1 in the Supplements,
respectively.
After impoundment, several different types of islands formed in the reservoir (Fig.
S1). This study selected a typical island for investigation (182 m in length, 90 m in
width), which is located at the convex bank (24°43′44″ N, 100°23′5″ E) in the sidebay
of Manwan reservoir, 30 km away from the dam (Fig. 1). Manwan has a subtropical
plateau monsoon climate, featuring no distinct seasons. Under reservoir operation, the
island bank is frequently flooded (Fig. S2).
**2.2 Monitoring wells**
Ten monitoring wells were installed in the island bank at 0.5 (W1), 1.5 (W2), 3.5 (W3),



6.5 (W4), 10.5 (W5), 15.5 (W6) 20.5 (W7), 25.5 (W8), 30.5 (W9), and 35.5 m (W10)
from the waterline, respectively (Fig. S2). The wells were 90-mm diameter perforated
polyvinylchloride pipes, reaching a depth of 2.0 m below the ground surface. To prevent
flooding, the wells were extended 2.0 m aboveground. Due to hydropower production,
the reservoir runs in a pseudo-periodic hydrological regime with cyclic water level
fluctuations. Here, we monitored a complete cycle of water level fluctuation within 115
h. Water levels were measured every 10 min from 11 to 16 September 2016 using
automated water level recorders (U2000101, OneSetHoBo, USA), which were mounted
at the bottom of W5, W7, W10, and the reservoir (Fig. S3).
Instantaneous lateral fluid fluxes ($q$) across the island bank per unit length were
calculated following the Darcy Eq. (1) (Gerecht et al., 2011; Hucks Sawyer et al., 2009):
$$q(t) = -Kb \cdot \left[\frac{\vartheta h(x,t)}{\vartheta x}\right] \tag{1}$$
where $Kb$ is sediment transmissivity, m d$^{-1}$; $h$ is hydraulic head, m; $x$ is distance, m; and
$t$ is time, d. A positive $q$ value indicates flow from the reservoir to the island. The island
$Kb$ was 0.99 m d$^{-1}$, which was measured according to Philip (1993).
**2.3 Sampling and physicochemical analysis**
After water level receded at the monitoring time of 100 h, groundwater (100 ml) was
carefully sampled in triplicate from each monitoring well with a portable peristaltic
pump (SC-1/253Yx, Chongqing Jieheng Peristaltic Pump Co., Ltd., China), and then
filtered *in situ* using portable syringe filters for water DOC analysis. Sediment (5 g)
was synchronously collected in triplicate from 10 cm below the surface adjacent to each
well using a hand shovel, and then homogenized before the storage for the analyses of





sediment OC and microbe. At a reservoir site adjacent to W1, water and surface
sediment samples were also collected in triplicate using a stainless-steel bucket and an
Ekman grab sampler, respectively. The collected water and sediment samples were kept
frozen in an ice box (-5 ℃―10 ℃) and transported to the laboratory for analysis within
three days.
Water temperature (WT), dissolved oxygen (DO), pH, and electrical conductivity
(EC) at each well were measured *in situ* using a multi-sensor probe (YSI 6600, Yellow
Springs Instruments, USA). Analysis of dissolved organic carbon (DOC) in the water
was conducted on filtered samples (Whatman GF/F, UK) using a total organic carbon
analyzer (Liqui TOC II, Elementar Inc., Germany). Sediment OC was determined using
a vario MACRO cube elementar (Elementar Inc., Germany). Fresh sediment was
freeze-dried and ground before analysis. Approximately 30 mg of each sample was
weighed in a tin cup and acidified with two drops of 8 % $H_3PO_4$ to remove inorganic
carbonates before OC analysis.
**2.4 Methane flux analysis**
Methane fluxes from the reservoir (eight sampling sites) and island (seventeen sampling
sites) were analyzed using bifunctional chambers according to the static chamber
method (Duchemin et al., 1999). The sampling sites are shown in Fig. S4. The
plexiglass bifunctional chamber consisted of a 6.28-L cylinder (20 cm in diameter, 20
cm in height) and a removable Styrofoam collar. During gas collection in the reservoir,
the chamber was fitted with the Styrofoam collar, which maintained the upper closed
portion of the chamber about 10 cm above the water surface (Fig. S5). The chambers





were left to stand for 20 min before sample collection. Gas samples (20 ml) were
collected every 10 min over a 40-min period using a 25-ml polypropylene syringe and
injected into a pre-evacuated Exetainer® vial (839 W, Labco, UK) for storage until
analysis using a gas chromatograph (7890B, Agilent Technologies, USA). Gas fluxes
were calculated using linear regression based on the concentration changes of five
samples over time. Linear regression correlation coefficients of less than 0.95 were not
accepted for further calculations (Duchemin et al., 1999). Simple spline interpolation
was used to interpolate the methane emissions from the sampling sites into space in the
reservoir and island separately (Immerzeel et al., 2009). Methane emission areas at
eight different categories were also calculated in the island.
**2.5 Microbial abundance analysis**
After being transported to the laboratory, the frozen sediment samples were stored
immediately at -80 ℃ for further molecular analysis. The sediment methanogens and
methanotrophs adjacent to each monitoring well across the island (ten sediment samples)
were quantified using qPCR. DNA extraction was undertaken using a FastDNA Power-
Max Soil DNA Isolation Kit (MP Biomedical, USA) according to the manufacturer's
instructions. The qPCR assay was performed using primers targeting methanogenic
archaeal 16S rDNA (primer set, 1106F/1378R) and methanotrophic *pmoA* genes
(primer set, A189F/M661R) (Watanabe et al., 2007; Ma and Lu, 2011). Gene copies
were amplified and quantified in a Bio-Rad cycler equipped with the iQ5 real-time
fluorescence detection system and software (version 2.0, Bio-Rad, USA). All reactions
were completed in a total volume of 20 μL containing 10 μL SYBR® *Premix Ex Taq^{TM}*



(Toyobo, Japan), 0.5 mM of each primer, 0.8 µL of BSA (3 mg mL$^{-1}$, Sigma, USA),
ddH$_2$O, and template DNA. The qPCR program for archaeal 16S rDNA was as follows:
℃ for 60 s, followed by 40 cycles of 95 ℃ for 25 s, 57 ℃ for 30 s, and 72 ℃ for
60 s. The qPCR program for *pomA* commenced with 95 ℃ for 60 s, followed by 40
cycles of 95 ℃ for 25 s, 53 ℃ for 30 s, and 72 ℃ for 60 s. A standard curve was
established by serial dilution ($10^{-2}$–$10^{-8}$) of known concentration plasmid DNA with the
target fragment. All PCRs were run in triplicate on 96-well plates (Bio-Rad, USA)
sealed with optical-quality sealing tape (Bio-Rad, USA). Three negative controls
without the DNA template were included for each PCR run.
**2.6 Data analysis**
One-way analysis of variance (ANOVA) was employed to test the statistical
significance of differences between sampling sites. Post-hoc multiple comparisons of
treatment means were performed using the Tukey's least significant difference
procedure. All statistical calculations were performed using the SPSS (v22.0) statistical
package for personal computers. The level of significance was $P < 0.05$ for all tests.
**3 Results**
**3.1 Physicochemical characteristics**
As shown in Fig. 2, the island groundwater had lower DO and pH, but higher WT, EC,
and DOC, compared with that of the bulk reservoir water. Lateral gradients of
groundwater pH and DO, and DOC were observed in the island. From the island edge
to the center, pH gradually increased from 6.55 ±0.13 to 7.25 ±0.12, whereas DO and
DOC decreased significantly from 4.80 ±0.19 to 0.43 ±0.09 mg L$^{-1}$ and 7.30 ±0.54 to

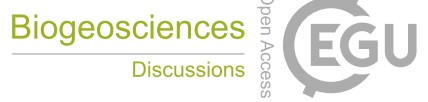



1.70 $\pm$ 0.39 mg L$^{-1}$, respectively ($P < 0.05$). There were no significant differences in
WT or DO between sampling sites ($P > 0.05$), which ranged from 15.9–17.4 ℃ and
390–761 µS cm$^{-1}$, respectively (Fig. 2a-e). In general, sediment OC was higher near the
island edge, decreasing from 6.37 $\pm$ 0.69 mg g$^{-1}$ at the edge to 2.42 $\pm$ 0.60 mg g$^{-1}$ at the
center of the island. Sediment OC in the reservoir was 6.63 $\pm$ 0.09 mg g$^{-1}$ (Fig. 2e).
**3.2 Water level fluctuation and hyporheic exchange**
The reservoir stage fluctuated frequently during the field survey, showing three distinct
peaks, with a maximum of 3.80 m in the first 37 h and gradual decline to below 1.30 m
in the next 60 h, yielding a maximum oscillation of 2.54 m. Similar oscillations were
observed in the island water table, but were damped and lagged relatively to the
reservoir stage fluctuations (Fig. 3a). In W5, W7, and W10, the water levels reached
3.27, 3.41, and 3.33 m, then fell to 1.74, 2.09, and 2.01 m, for a maximum oscillation
of 1.53, 1.33, and 1.32 m, respectively. Data from the automated water level recorders
indicated that the water level responses in W5, W7, and W10 lagged the reservoir stage
by 20, 25, and 30 min, respectively. Lateral hyporheic exchanges across the island bank
were calculated according to the Darcy Law, showing that the flux was largest at the
island edge and decreased from the edge to the center. The water exchange across the
0–10.5 m island edge zone was 1.2 and 4.7 times higher than those across the 10.5–20.5
m and 20.5–35.5 m zones, respectively. The flow rates at the reservoir-W5, W5-W7,
and W7-W10 zones were relatively consistent at -0.55–1.35, -0.89–0.28, and -0.39–
0.17 m$^2$ d$^{-1}$ (Fig. 3b), resulting in a water exchange volume of 2.61, 2.26, and 0.56 m$^3$,
respectively, over the 115-h observation period.





### 3.3 Methane emissions

High methane emission rates were observed at the island sites, with a maximum of 10.4

mg h$^{-1}$m$^{-2}$ at the center. However, a large ring-like low methane emission zone appeared

around the island edge, where the methane flux was maintained at -0.2–1.6 mg h$^{-1}$m$^{-2}$

(Fig. 4a). The negative flux values also suggest the occurrence of a methane sink at the

island edge. The ring-like zone accounted for 89.1 % of the island area, of which 9.1 %

accounted for the methane sink zone (Fig. 4b). Compared with the island, the methane

flux from the adjacent reservoir was moderate at 0.4–5.5 mg h$^{-1}$m$^{-2}$ (Fig. 4a).

### 3.4 Methanogen and methanotroph abundances

Methanogens and methanotrophs were distributed non-uniformly across the island. In

general, methanogen counts were low at the island edge but high at the center, whereas

methanotrophs were abundant at the island edge but scarce in the center. From the island

edge to the center, the methanogenic archaeal 16S rDNA gene increased from $0.12 \times$

$10^5$ to $5.34 \times 10^5$ copies g$^{-1}$, and the methanotrophic *pmoA* gene decreased from $1.57 \times$

$10^6$ to $0.64 \times 10^6$ copies g$^{-1}$.

### 4 Discussion

### 4.1 Hyporheic exchange and redox gradients

The hyporheic zone is the interface beneath and adjacent to streams and rivers, where

water, heat, nutrients and contaminants are exchanged and many biogeochemical

reactions occur (Cardenas and Markowski, 2010; Tonina and Buffington, 2011). In

hydropower reservoirs, the release of water pulses is often employed to increase power

production and meet daily electricity peak demand (Bonalumi et al., 2012; Toffolon et



al., 2010). Such hydropeaking creates daily water level fluctuations in the reservoir. In
this study, frequent water level fluctuations were observed within the 115-h observation
period, with a maximum of 3.80 m (Fig. 3a). A hysteretic response occurred in the
island bank water table (Fig. 3a), driving water exchange between the reservoir and
island (Fig. 3b). The water exchange flux was largest close to the island edge and
decreased from the edge to the center, as water table fluctuations were attenuated (Fig.
3a).

During a storage-release cycle, the island switched from water gaining to losing at

daily or hourly scales, creating a ring-like zone of enhanced hyporheic exchange around
the island. The hyporheic zone extended tens of meters into the island bank (Fig. 3b).
If the river system was unregulated, however, hydrodynamics within the hyporheic zone
would likely exhibit seasonal or annual patterns, or keep pace with snowmelt and
rainstorm events, under a natural base flow-fed regime. In this case, hyporheic zones
may be limited or altogether absent (Boano et al., 2008; Cardenas and Wilson, 2007).
Exchange across the sediment-water interface involves mixing of surface water and
groundwater through hyporheic flow (Hester et al., 2013; Naranjo et al., 2015). In this
study, when the reservoir water entered the hyporheic flow path, it was typically rich in
oxygen (Fig. 2d). As oxygen was consumed through aerobic respiration, other terminal
electron acceptors were utilized (Klupfel et al., 2014), creating a redox gradient along
the hyporheic flow path (Fig. 2d). Changes in sediment moisture can speed up the
mineralization of organic matter (Wang et al., 2010; Rubol et al., 2014). Groundwater
DOC showed a general decrease from the island edge to center (Fig. 2e). This hyporheic



exchange clearly affected biogeochemical processes, and had important effects on
hyporheic microbial communities, especially redox-sensitive species. For example, we
detected poor methanogen abundance at the island edge, but rich abundance at the
center, with methanotrophs showing the opposite pattern (Fig. 5). The spatial
heterogeneity of sediment OC in the island was likely due to the settled particles with
different organic matter contents during the island formation, or the release of the
exudates from benthic biofilms, including algae and other primary producers, under
hyporheic exchanges (Rubol et al., 2014).

**4.2 Self-mitigation of methane emissions**

Methane is the second most important greenhouse gas, contributing approximately 18 %
to total global warming effects (Smith et al., 2013; Wuebbles and Hayhoe, 2002). Inland
waters (lakes, rivers, and reservoirs) are significant sources of atmospheric methane,
which is mainly released from anoxic sediment (Bastviken et al., 2011; Sobek et al.,
2012). In dammed rivers, riverbed sediment accumulation in forebays and sidebay
islands creates potential methane emission hotspots. In this study, however, high
methane emissions were only observed at the island center, with a ring-like low
methane emission zone or even methane sink appearing around the island edge (Fig.
4a). In natural waters, methane is primarily produced by methanogens under anaerobic
conditions (Yang et al., 2017). Along the redox gradient in the island (Fig. 2d), methane
production was inhibited at the edge and favored at the center, as indicated by the lower
abundance of sediment methanogens at the island edge than at the center (Fig. 5). The
methane sink at the island edge was mainly attributed to oxidation consumption by



methanotrophs. The aerobic sediment at the island edge was rich in methanotrophs (Fig.
5), which may consume methane to below equilibrium with the atmosphere, driving a
net air-water flux. Groundwater DOC and sediment OC at the island edge, which are
carbon sources for methane emission, were higher than that at the island center (Fig.
2e,f), suggesting that both sediment heterogeneity and dilution effects of hyporheic
exchange had limited contribution to the spatial pattern of methane emissions in the
island. Hyporheic exchange effectively shifted redox gradients across the island,
resulting in substantial mitigation of potential methane emissions. In this study, only
0.2 % of the island area maintained a high methane flux (9.6–11.2 mg h$^{-1}$m$^{-2}$) (Fig. 4b),
suggesting that methane emissions across the small island were attenuated, but only in
the area where hyporheic exchange occurred. It may be possible for methane emission
hotspots in larger islands to be mitigated by enlarging their hyporheic zone. For
example, artificial channels can be made through the island to create a hyporheic zone
in the center (Fig. S6). In addition, the maximum water level in the reservoir can be
raised by modifying hydropower operation scenarios to extend the hyporheic zone (Fig.
S6).
**4.3 Implications**
Greenhouse gas emissions significantly detract from the green credentials of
hydropower, and have thus received considerable research attention (Giles, 2006; Hu
and Cheng, 2013). Previous studies have revealed that damming causes significant
retention of carbon and creates deep, anoxic sediment strata, fueling methanogenesis
and net water-air methane flux (Maeck et al., 2013). In this study, the self-mitigation of



methane emissions was apparent in the hyporheic zone under the reservoir operations.
Given the widely distributed hyporheic zones in reservoirs, this self-mitigation should
be of concern in future estimations of greenhouse gas emissions from dammed rivers.
Prospective studies should assess the quantitative relationship between methane
emissions from the hyporheic zone and hydropower operation scenarios.

Until now, few studies have concentrated on organic carbon mineralization in the

hyporheic zone of reservoirs, with most focusing on the process of denitrification
(Zarnetske et al., 2011b). Carbon emissions in the hyporheic zone are poorly understood,
especially in regulated and dammed rivers. This study fills the knowledge gap and adds
to our understanding of the ecological impacts of hydropower exploitation. Under
reservoir operation, variable redox conditions and methane production may also affect
the mercury cycle in the hyporheic zone and thereby the release of methylmercury (a
bioaccumulative environmental toxicant) to the river (Marvin-DiPasquale et al., 2009),
a subject deserving of further study.

The methods used in this study had some limitations. First, an average value of

hydraulic conductivity was chosen for calculating the Darcy fluxes, which does not
reflect the full heterogeneity of island sediment, which ranges from silt and fine clay to
sand. Second, direct measurements in the open monitoring wells introduced
atmospheric oxygen into the previously isolated groundwater, presenting possible
systematic errors in the groundwater data. However, even with these potential
complications, the data obtained in the present study were useful for clarifying the
biogeochemical processes in the hyporheic zone associated with reservoir operation.



**5 Conclusions**


In dammed rivers, sediment deposited islands are widely distributed in sidebays and are
potential hotspots of methane emission to the atmosphere. In this study, high methane
fluxes were only observed at the island center, while a ring-like zone of low methane
emission or even sink was found around the island edge. We attribute this methane
mitigation to hyporheic exchange between the reservoir and island. Under reservoir
operation, frequent water level fluctuations drove hyporheic exchange, creating redox
gradients along the hyporheic flowpath. These redox gradients affected the microbial
communities associated with methane production and consumption, producing a net
effect of methane emission self-mitigation. Our understanding of this self-mitigation of
methane emission in dammed rivers will help us to screen effective strategies seeking
to lessen the global warming impacts of hydropower systems.

**Data availability**


The data presented here can be obtained upon request to Wenqing Shi (wqshi@nhri.cn).

**Author contribution**


Qiuwen Chen designed the research; Wenqing Shi, Yuyu Ji and Yuchen Chen performed
the research; Cheng Chen and Juhua Yu contributed new reagents/analytic tools;
Qiuwen Chen and Wenqing Shi analyzed the data; Jianyun Zhang and Bryce R. Van
Dam contributed significant discussions and inputs; Wenqing Shi and Qiuwen Chen
wrote the paper with input from all authors.





**Competing interests**
The authors declare that they have no conflict of interest.

**Acknowledgements**
Funding for this study was provided by the National Nature Science Foundation of
China (No. 91547206, 51425902, 51709181 and 51709182).



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




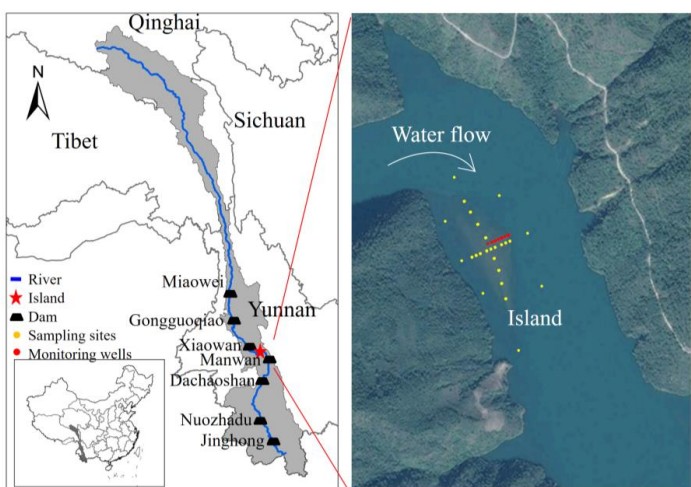


**Fig. 1** Map of the studied island in Manwan reservoir, Lancang-Mekong River.






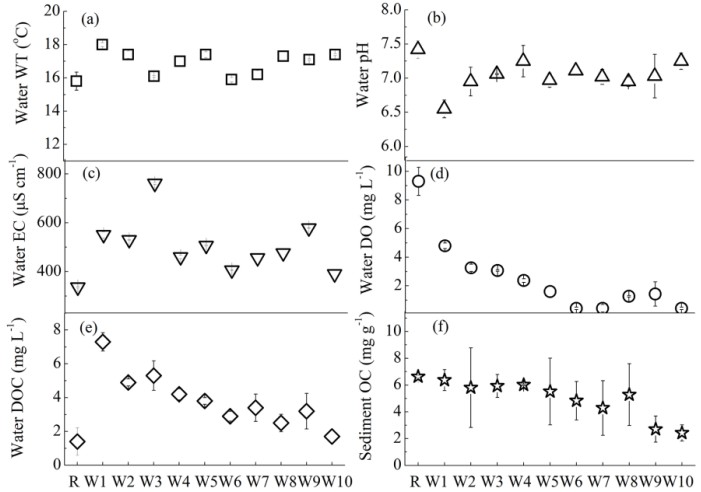

**Fig. 2** Physicochemical properties of the island and reservoir, including water (a) WT,
(b) pH, (c) EC, (d) DO, (e) DOC, and (f) sediment OC. Solid symbols represent the
physicochemical properties of the reservoir. WT = water temperature, EC = electrical
conductivity, DO = dissolved oxygen, DOC = dissolved oxygen carbon, OC = organic
carbon, R = reservoir, W = monitoring wells. Error bars indicate standard deviations.




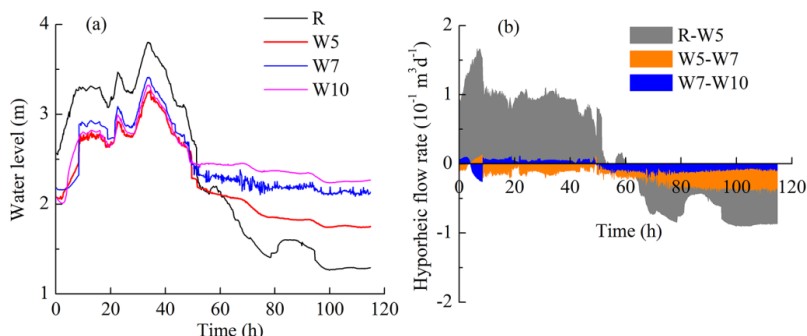


**Fig. 3** Vertical water level fluctuation (a) and horizontal hyporheic flow rate (b). R =

reservoir, W = monitoring wells. Positive fluxes indicate net flow from the reservoir to

island, whereas negative values indicate net flow from the island to reservoir.





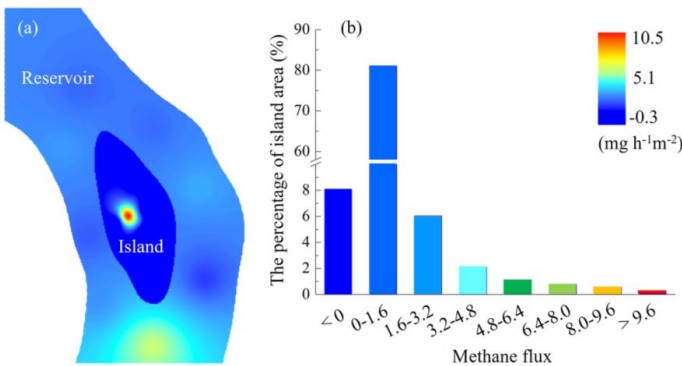


**Fig. 4** Methane emissions from the island and reservoir. (a) Spatial pattern of methane

emissions; (b) Percentage of the island area emitting methane at a certain flux. Methane

fluxes were interpolated separately for the island and reservoir.





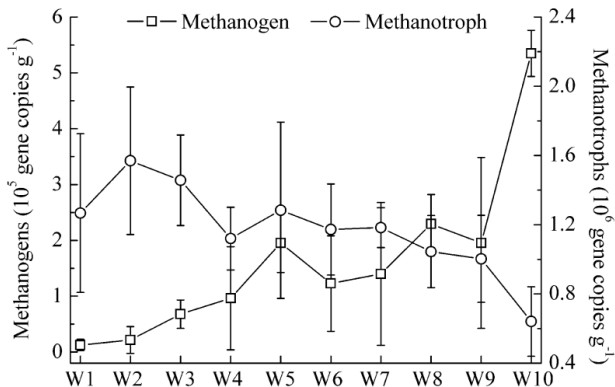


**Fig. 5** Abundances of sediment methanogens and methanotrophs in the island. W=
monitoring wells. Error bars indicate standard deviations.