# Peer review of "Fig. S1 The river impoundment formed islands in the Lancang-Mekong River. There were about 50 sediment-deposited islands, of which 54.2%, 12.5% and 33.3% were located at the convex bank (a), the concave bank (b), and the channel (c), respectively."

_Biogeosciences, 2018_

## Referee Comment (RC1) · Anonymous Referee #1 · 31 Oct 2018

The paper deals with CH4 dynamics in relation with methanotrophic and methanogenic bacteria abundance in the sediment/soils following topography from the shoreline to the top of an island located in a hydroelectric reservoir.

This is a very original study with unexpected outcomes: -negative CH4 fluxes at the water or sediment interface, never observed in aquatic systems and moist soils -maximum abundance of methanotrophs in the water saturated and organic soils/sediments of the shoreline, an environment supposed to be ideal for methanogens –maximum abundance of methanogens at the top of the highland which is more often uncovered with water and which therefore might favor oxygen diffusion and occurrence of methanotrophs

Major comments

[Figure]

-In the community studying greenhouse gas emissions from hydroelectric reservoir, the littoral zone and the area undergoing the water level variations is called the drawdown area. Some reference listed at the end of the review might be considered for improving the description of the research context of this study and some comparison with those results could be interesting and more papers can be found on this topic

-The sketch of the chamber in Supplementary material (S.5) did not show any vent for preventing an increase of the pressure in the chamber during the installation resulting from the decrease of the chamber volume when the edge of the chamber penetrates either in the water or the soils/sediment. Overpressure generate bias to the flux measurement by increasing the solubility of CH4 in the surface water or interstitial water. Such phenomenon could explain some of the negative fluxes and might have influence all CH4 fluxes measurements.

-No information is given in the manuscript for air-soil flux measurements (L148). Again, the design of the chamber calls into doubt the validity of the soil fluxes (See supplementary material S.5)). Usually, soil flux measurements are performed with chambers with collar which allow the installation of the collar a few hours before the flux measurement in order to avoid measuring a flux that might mostly result from the perturbation associated with the penetration of the chamber or collar edges in the soil/sediment. This might have increased significantly the soil fluxes.

-Sediment sampling strategy (L116) is not ideal if one wants to link fluxes at the air-water interface or air soil interface with bacterial abundance and functions. Fluxes are controlled by the balance between methanogenesis and methanotroph, the former being mostly active in "deep" anoxic horizons (10 cm in soils, sediments...) and the latter above the oxic-anoxic interface (typically in the first mm at the air soil interface). Therefore, a bulk sample of the first ten cm in the sediment might fail at describing the expected vertical structure of the bacterial community involved in the CH4 cycle. Cores might have been more adapted

-16S rDNA can be used for the detection of active bacteria and biogeochemical pathways in combination with isotopic labelling. Without labelling, DNA prove the presence of the bacteria (since it is very stable in natural environment) but it does not clearly demonstrate they are active. RNA might have been more adapted because of its shorter life time. Relationships between the CH4 fluxes and the bacterial abundance based on DNA must therefore be considered with care.

According to the doubts on the methodology and the sampling strategy, it is recommended to the authors to clarify those points before the quality of the paper can be fully evaluated

REF: Chen, H., X. Yuan, Z. Chen, Y. Wu, X. Liu, D. Zhu, N. Wu, Q. a. Zhu, C. Peng and W. Li (2011). "Methane emissions from the surface of the Three Gorges Reservoir." J. Geophys. Res. 116(D21): D21306.

Chen, H., Y. Wu, X. Yuan, Y. Gao, N. Wu and D. Zhu (2009). "Methane emissions from newly created marshes in the drawdown area of the Three Gorges Reservoir." J. Geophys. Res. 114: D18301.

Li, Z., Z. Zhang, C. Lin, Y. Chen, A. Wen and F. Fang (2016). "Soil–air greenhouse gas fluxes influenced by farming practices in reservoir drawdown area: A case at the Three Gorges Reservoir in China." Journal of Environmental Management 181: 64-73.

Serça, D., C. Deshmukh, S. Pighini, P. Oudone, A. Vongkhamsao, P. Guédant, W. Rode, A. Godon, V. Chanudet, S. Descloux and F. Guérin (2016). "Nam Theun 2 Reservoir four years after commissioning: significance of drawdown methane emissions and other pathways." Hydroécol. Appl. 19: 119-146.

Harrison, J. A., B. R. Deemer, M. K. Birchfield and M. T. O'Malley (2017). "Reservoir Water-Level Drawdowns Accelerate and Amplify Methane Emission." Environmental Science & Technology 51(3): 1267-1277.

Lu, F., L. Yang, X. Wang, X. Duan, Y. Mu, W. Song, F. Zheng, J. Niu, L. Tong, H. Zheng,

Y. Zhou, J. Qiu and Z. Ouyang (2011). "Preliminary report on methane emissions from the Three Gorges Reservoir in the summer drainage period." Journal of Environmental Sciences 23(12): 2029-2033.

---

## Referee Comment (RC2) · Anonymous Referee #2 · 1 Nov 2018

The manuscript presents an interesting strategy to estimate the spatial contribution of CH4 emissions from a sediment-deposited island in the Lancang-Mekong reservoir. While most of the studies in reservoirs are focused in estimate CH4 emissions in water sites, this study calls the attention to consider the sediment-island CH4 dynamics during reservoir operations (water level fluctuations). I think the study is relevant as strategy in such ecosystems (sediment-deposited island), but there are major clarifications that should be addressed before going a step forward in the acceptance.

Microbial analysis issues.

Due to the methodology of sampling (no with a core system) and different qPCR programs (I understand but you need explain it); methanotrophy and methanogenesis statements are overstated in the whole manuscript. I would recommend focusing more

into the water level fluctuation to explain the CH4 dynamics obtained. Additionally, Figure 5 contains an artifact in the axis to pronounce your statements in the manuscript, please use the same axis level to avoid confusion, even a statistical analysis comparing each site (methanogens vs methanotrophs) and whole samples may help to approve such patterns.

CH4 measurements.

While, the title of the manuscript is about CH4 emission. You have only one figure (Figure 4), and a small result description from the same figure (section 3.3) about CH4 emission patterns. Moreover, Figure 4 is overinterpreted by the extrapolation, which is valid, but you should explain the range of uncertain.

I have seen CH4 consumption in flux measurements in soils, but in water determinations are very strange. If you are confident of your measurements, I encourage you to include the data obtained, and explore relations with other parameters measured in the study.

Did you sample only once per time in each chamber? If you did it, you have large uncertain in your values, since there are a lot of risk to have leaks and lost sample before determining in the gas chromatograph. Which was the volume of gas pre-evacuated in each vial?

General statements.

There are several statements that you are not able to support with your data, it would help you make a list of different parameters obtained and see how to relate them. Then, avoid such interpretation about redox changes, exudates, among others.

Sections 4.2 and 4.3 contains several discussions unnecessary (they are more introduction than discussion) about the relevance of the CH4 emission from aquatic ecosystems, how methane is produced biologically, the environment relevance of reservoirs, shelf mitigation, limitations using Darcy fluxes.

---

## Author Comment (AC1) · 20 Nov 2018

We are indeed grateful to the referee for the positive comments and valuable suggestions that have enabled us to improve the manuscript. We seriously considered all the comments and made a thorough revision. The changes made in the text were shown with a red color font. Detailed point-to-point responses and the revised manuscript are listed in the supplement.

Please also note the supplement to this comment:
https://www.biogeosciences-discuss.net/bg-2018-380/bg-2018-380-AC1-supplement.pdf

---

## Author Comment (AC2) · 20 Nov 2018

**Referee # 1**

The paper deals with $CH_4$ dynamics in relation with methanotrophic and methanogenic bacteria abundance in the sediment/soils following topography from the shore line to the top of an island located in a hydroelectric reservoir.

This is a very original study with unexpected outcomes: -negative $CH_4$ fluxes at the water or sediment interface, never observed in aquatic systems and moist soils-maximum abundance of methanotrophs in the water saturated and organic soils/sediments of the shoreline, an environment supposed to be ideal for methanogens –maximum abundance of methanogens at the top of the highland which is more often uncovered with water and which therefore might favor oxygen diffusion and occurrence of methanotrophs.

**Major comments**

(1) In the community studying greenhouse gas emissions from hydroelectric reservoir, the littoral zone and the area undergoing the water level variations is called the drawdown area. Some reference listed at the end of the review might be considered for improving the description of the research context of this study and some comparison with those results could be interesting and more papers can be found on this topic.

- Many thanks for the valuable suggestions. We have carefully studied the listed references and improved the description of the research context of this study. Please see line19, 21 on page 2, line 57, 66, 67 on page 4, line 222 on page 11, line 249 on page 12, line 250, 252, 254 on page 13, line 288 on page 14, line 298, 303, 305, 306, 310 on page 15, line 318 on page 16 in the revised manuscript.

- According to the suggestions, we have added the comparison between the results of this study with others on this topic. Please see line 295-300 on page 15 in the revised manuscript.

(2) The sketch of the chamber in Supplementary material (S.5) did not show any vent for preventing an increase of the pressure in the chamber during the installation resulting from the decrease of the chamber volume when the edge of the chamber penetrates either in the water or the soils/sediment. Overpressure generate bias to the flux measurement by increasing the solubility of $CH_4$ in the surface water or interstitial water. Such phenomenon could explain some of the negative fluxes and might have influence all $CH_4$ fluxes measurements.

- During the chamber installation, the outlet of the chamber was open (**Fig. 1**), and was left to stand for 20 min to equilibrate with ambient pressure outside. Then, a drop of water was filled into the outlet to test whether the pressure inside was equilibrated or not before the outlet was closed. Based on our results of previous chambers (Fig. 2), we did not design another outlet for equilibrating pressure in order to increase the gas tightness of this chamber. We have updated the descriptions to make it clearer, and please see line 145-147 on page 8 in the revised manuscript.

[Figure]

**Fig. 1** The gas collection chamber for gas fluxes across the air-water interfaces in this study

[Figure]

**Fig. 2** The gas collection chamber for gas fluxes across the air-water interfaces in our previous studies (3) No information is given in the manuscript for air-soil flux measurements (L148). Again, the design of the chamber calls into doubt the validity of the soil fluxes (See supplementary material S.5)). Usually, soil flux measurements are performed with chambers with collar, which allows the installation of the collar a few hours before the flux measurement in order to avoid measuring a flux that might mostly result from the perturbation associated with the penetration of the chamber or collar edges in the soil/sediment. This might have increased significantly the soil fluxes.

- We are really sorry for having not described the air-soil flux measurements. We fully agree with the referee that soil flux measurements should be performed with chambers with collar. We had planned to set up perpetual collars in the study area; however, the frequent water level fluctuation often filled the collar with water in our preliminary tests. According to Schindlbacher et al. (2012), the chamber was carefully inserted 5 cm deep into the sediment, leaving 15 cm above the sediment surface. The outlet of the chamber was open during the chamber deployment, and was left to stand for 90 min to equilibrate before sample collection. The information has been updated in line 147-151 on page 8 in the revised manuscript. For the accurate estimation of $CH_4$ emissions in this special area, further studies are needed in future using the method of noncontact measurements, such as Eddy Covariance Measurement Systems.

Schindlbacher, A., Wunderlich, S., Borken, W., Kitzler, B., Zechmeister‐Boltenstern, S., & Jandl, R. (2012). Soil respiration under climate change: prolonged summer drought offsets soil warming effects. *Global Change Biology*, 18(7), 2270-2279.

(4) Sediment sampling strategy (L116) is not ideal if one wants to link fluxes at the air-water interface or air soil interface with bacterial abundance and functions. Fluxes are controlled by the balance between methanogenesis and methanotroph, the former being mostly active in "deep" anoxic horizons (10 cm in soils, sediments...) and the latter above the oxic-anoxic interface (typically in the first mm at the air soil interface). Therefore, a bulk sample of the first ten cm in the sediment might fail at describing the expected vertical structure of the bacterial community involved in the $CH_4$ cycle. Cores might have been more adapted

- Yes, in natural aquatic ecosystems, methanogenesis is mostly active in "deep" anoxic horizons (10 cm in soils, sediments...) and methanotroph mainly occurs above the oxic-anoxic interface. In this study, frequent reservoir operation induced lateral hyporheic exchange and created oxygen gradients along the hyporheic flow path across the island (Fig. 3), regulating microbes for $CH_4$ emissions. Our interests focus on the horizontal heterogeneity of microbial abundance here to clarify spatial patterns of $CH_4$ emission. Sediment was collected from 10 cm below the surface in order to avoid the disturbances of oxygen penetration from air. For the vertical heterogeneity of bacterial community structure, we can design and conduct further studies using a core system according to the referee's suggestions.

[Figure]

**Fig. 3** The horizontal heterogeneity in the interior of island under reservoir operation.

(5) 16S rDNA can be used for the detection of active bacteria and biogeochemical pathways in combination with isotopic labelling. Without labelling, DNA prove the presence of the bacteria (since it is very stable in natural environment) but it does not clearly demonstrate they are active. RNA might have been more adapted because of its shorter life time. Relationships between the $CH_4$ fluxes and the bacterial abundance based on DNA must therefore be considered with care.

- Thanks for the valuable comments. Methane emissions may not only rely on bacterial abundance but also bacterial activity (Schwarz et al., 2008). Some other molecular biology techniques need to be used in future studies according to the referee's suggestions. The information has been updated to make the discussion appropriate in this study. Please refer to line 277-280 on page 14 in the revised manuscript.

Schwarz, J. I. K., Eckert, W., Conrad, R. Response of the methanogenic microbial community of a profundal lake sediment (Lake Kinneret, Israel) to algal deposition. *Limnology and Oceanography*, 2008, 53(1), 113-121.

REF:

Chen, H., X. Yuan, Z. Chen, Y. Wu, X. Liu, D. Zhu, N. Wu, Q. a. Zhu, C. Peng and W. Li (2011). "Methane emissions from the surface of the Three Gorges Reservoir." J. Geophys. Res. 116(D21): D21306.

Chen, H., Y. Wu, X. Yuan, Y. Gao, N. Wu and D. Zhu (2009). "Methane emissions from newly created marshes in the drawdown area of the Three Gorges Reservoir." J. Geophys. Res. 114: D18301.

Li, Z., Z. Zhang, C. Lin, Y. Chen, A. Wen and F. Fang (2016). "Soil–air greenhouse gas fluxes influenced by farming practices in reservoir drawdown area: A case at the Three Gorges Reservoir in China." Journal of Environmental Management 181: 64-73.

Serça, D., C. Deshmukh, S. Pighini, P. Oudone, A. Vongkhamsao, P. Guédant, W. Rode, A. Godon, V. Chanudet, S. Descloux and F. Guérin (2016). "Nam Theun 2 Reservoir four years after commissioning: significance of drawdown methane emissions and other pathways." Hydroécol. Appl. 19: 119-146.

Harrison, J. A., B. R. Deemer, M. K. Birchfield and M. T. O'Malley (2017). "Reservoir Water-Level Drawdowns Accelerate and Amplify Methane Emission." Environmental Science & Technology 51(3): 1267-1277.

Lu, F., L. Yang, X. Wang, X. Duan, Y. Mu, W. Song, F. Zheng, J. Niu, L. Tong, H. Zheng, Y. Zhou, J. Qiu and Z. Ouyang (2011). "Preliminary report on methane emissions from the Three Gorges Reservoir in the summer drainage period." Journal of Environmental Sciences 23(12): 2029-2033.

[revised manuscript text omitted]
 outlet of the chamber was open during the chamber deployment, and was left to stand for 20 min to equilibrate with ambient pressure outside before sample collection. During gas collection on the island, the chamber was carefully inserted 5 cm deep into the sediment, leaving 15 cm above the sediment surface. The outlet of the chamber was also open during the chamber deployment, and was left to stand for 90 min to equilibrate before sample collection. Gas samples (20 ml) were collected every 10 min over a 40-min period using a 25-ml polypropylene syringe and injected into a 12-ml pre-evacuated

Exetainer® vial (839 W, Labco, UK) for storage until analysis using a gas chromatograph (7890B, Agilent Technologies, USA). Gas fluxes were calculated using linear regression based on the concentration changes of five samples over time. Linear regression correlation coefficients of less than 0.95 were not accepted for further calculations (Duchemin et al., 1999). Methane fluxes at each site were measured in triplicate by placing three individual chambers. Simple spline interpolation was used to interpolate the methane emissions from the sampling sites into space in the reservoir and island separately (Immerzeel et al., 2009), and the range of the uncertain was 0.05

mg h$^{-1}$ m$^{-2}$. Methane emission areas at eight different categories were also calculated in the island.

**2.5 Microbial abundance analysis**

After being transported to the laboratory, the frozen sediment samples were stored immediately at -80 ℃ for further molecular analysis. The sediment methanogens and methanotrophs adjacent to each monitoring well across the island (ten sediment samples)

were quantified using qPCR. DNA extraction was undertaken using a FastDNA Power-

Max Soil DNA Isolation Kit (MP Biomedical, USA) according to the manufacturer's instructions. The qPCR assay was performed using primers targeting methanogenic archaeal 16S rDNA (primer set, 1106F/1378R) and methanotrophic *pmoA* genes (primer set, A189F/M661R) (Watanabe et al., 2007; Ma and Lu, 2011). Gene copies were amplified and quantified in a Bio-Rad cycler equipped with the iQ5 real-time fluorescence detection system and software (version 2.0, Bio-Rad, USA). All reactions were completed in a total volume of 20 μL containing 10 μL SYBR® *Premix Ex Taq*$^{TM}$

(Toyobo, Japan), 0.5 mM of each primer, 0.8 μL of BSA (3 mg mL$^{-1}$, Sigma, USA), ddH$_2$O, and template DNA. The qPCR program mainly depends on the sequence and length of functional genes and primers used, and different qPCR programs for archaeal

16S rDNA and *pomA* were applied in this study. The qPCR program for archaeal 16S

rDNA was as follows: 95 ℃ for 60 s, followed by 40 cycles of 95 ℃ for 25 s, 57 ℃

for 30 s, and 72 ℃ for 60 s, and the qPCR program for *pomA* referred to: 95 ℃ for 60

s, followed by 40 cycles of 95 ℃ for 25 s, 53 ℃ for 30 s, and 72 ℃ for 60 s. A standard curve was established by serial dilution ($10^{-2}$–$10^{-8}$) of known concentration plasmid

DNA with the target fragment. All PCRs were run in triplicate on 96-well plates (Bio-

Rad, USA) sealed with optical-quality sealing tape (Bio-Rad, USA). Three negative controls without the DNA template were included for each PCR run.

**2.6 Data analysis**

One-way analysis of variance (ANOVA) was employed to test the statistical significance of differences between sampling sites. Post-hoc multiple comparisons of treatment means were performed using the Tukey's least significant difference procedure. All statistical calculations were performed using the SPSS (v22.0) statistical package for personal computers. The level of significance was $P < 0.05$ for all tests.

**3 Results**

**3.1 Physicochemical characteristics**

As shown in Fig. 2, the island groundwater had lower DO but higher DOC, compared with that of the bulk reservoir water. Lateral gradients of groundwater DO and DOC

were observed in the island. From the island edge to the center, DO and DOC decreased significantly from $4.80 \pm 0.19$ to $0.43 \pm 0.09$ mg L$^{-1}$ and $7.30 \pm 0.54$ to $1.70 \pm 0.39$ mg

L$^{-1}$, respectively ($P < 0.05$) (Fig. 2a,b). In general, sediment OC was higher near the island edge, decreasing from $6.37 \pm 0.69$ mg g$^{-1}$ at the edge to $2.42 \pm 0.60$ mg g$^{-1}$ at the center of the island. Sediment OC in the reservoir was $6.63 \pm 0.09$ mg g$^{-1}$ (Fig. 2c).

**3.2 Water level fluctuation and hyporheic exchange**

The reservoir stage fluctuated frequently during the field survey, showing three distinct peaks, with a maximum of 3.80 m in the first 37 h and gradual decline to below 1.30 m in the next 60 h, yielding a maximum oscillation of 2.54 m. Similar oscillations were observed in the island water table, but were damped and lagged relatively to the reservoir stage fluctuations (Fig. 3a). In W5, W7, and W10, the water levels reached 3.27, 3.41, and 3.33 m, then fell to 1.74, 2.09, and 2.01 m, for a maximum oscillation of 1.53, 1.33, and 1.32 m, respectively. Data from the automated water level recorders indicated that the water level responses in W5, W7, and W10 lagged the reservoir stage by 20, 25, and 30 min, respectively. Lateral hyporheic exchanges across the island bank were calculated according to the Darcy Law, showing that the flux was largest at the island edge and decreased from the edge to the center. The water exchange across the 0–10.5 m island edge zone was 1.2 and 4.7 times higher than those across the 10.5–20.5 m and 20.5–35.5 m zones, respectively. The flow rates at the reservoir-W5, W5-W7, and W7-W10 zones were relatively consistent at -0.55–1.35, -0.89–0.28, and -0.39–0.17 $m^2$ $d^{-1}$ (Fig. 3b), resulting in a water exchange volume of 2.61, 2.26, and 0.56 $m^3$, respectively, over the 115-h observation period.

**3.3 Methane emissions**

High methane emission rates were observed at the island sites, with a maximum of 10.4 mg $h^{-1}m^{-2}$ at the center. However, a large ring-like low methane emission zone appeared at the drawdown area around the island edge, where the methane flux was maintained at -0.2–1.6 mg $h^{-1}m^{-2}$ (Fig. 4a). The negative flux values also suggest the occurrence of a methane sink at the island edge. The ring-like zone accounted for 89.1 % of the island area, of which 9.1 % accounted for the methane sink zone (Fig. 4b). Compared with the island, the methane flux from the adjacent reservoir was moderate at 0.4–5.5 mg $h^{-1}m^{-2}$ (Fig. 4a).

**3.4 Methanogen and methanotroph abundances**

Methanogens and methanotrophs were distributed non-uniformly across the island. In general, methanogen counts were low at the island edge but high at the center, whereas methanotrophs were abundant at the island edge but scarce in the center. From the island edge to the center, the methanogenic archaeal 16S rDNA gene increased from $0.12 \times 10^5$ to $5.34 \times 10^5$ copies $g^{-1}$, and the methanotrophic *pmoA* gene decreased from $1.57 \times 10^6$ to $0.64 \times 10^6$ copies $g^{-1}$ (Fig. 5a). The ratio of methanogen to methanotroph abundance increased from 0.01 at the island edge to 0.83 at the center (Fig. 5b).

**4 Discussion**

**4.1 Hyporheic exchange and oxygen gradients**

In hydropower reservoirs, the release of water pulses is often employed to increase power production and meet daily electricity peak demand (Bonalumi et al., 2012; Toffolon et al., 2010). Such hydropeaking creates daily water level fluctuations in the reservoir. In this study, frequent water level fluctuations were observed within the 115-h observation period, with a maximum of 3.80 m (Fig. 3a). A hysteretic response occurred in the island bank water table (Fig. 3a), driving water exchange between the reservoir and island (Fig. 3b). The water exchange flux was largest close to the island edge and decreased from the edge to the center, as water table fluctuations were attenuated (Fig. 3a).

During a storage-release cycle, the island switched from water gaining to losing at daily or hourly scales, creating a ring-like drawdown area of enhanced hyporheic exchange around the island. The drawdown area extended tens of meters into the island bank (Fig. 3b). If the river system was unregulated, however, hydrodynamics within the drawdown area would likely exhibit seasonal or annual patterns, or keep pace with snowmelt and rainstorm events, under a natural base flow-fed regime. In this case, the drawdown area may be limited or altogether absent (Boano et al., 2008; Cardenas and

Wilson, 2007).

Exchange across the sediment-water interface involves mixing of surface water and groundwater through hyporheic flow (Hester et al., 2013; Naranjo et al., 2015). In this study, when the reservoir water entered the hyporheic flow path, it was typically rich in oxygen (Fig. 2d). As oxygen was consumed through aerobic respiration, other terminal electron acceptors were utilized (Klupfel et al., 2014), creating an oxygen gradient along the hyporheic flow path (Fig. 2d). Changes in sediment moisture can speed up the mineralization of organic matter (Wang et al., 2010; Rubol et al., 2014).

Groundwater DOC showed a general decrease from the island edge to center (Fig. 2e).

This hyporheic exchange clearly affected biogeochemical processes, and had important effects on hyporheic microbial communities, especially oxygen-sensitive species. For example, we detected poor methanogen abundance at the island edge, but rich abundance at the center, with methanotrophs showing the opposite pattern (Fig. 5).

**4.2 Spatial heterogeneity of methane emissions**

In dammed rivers, riverbed sediment accumulation creates potential methane emission hotspots. In this study, however, high methane emissions were only observed at the island center, with a ring-like low methane emission zone or even methane sink appearing around the island edge (Fig. 4a). This was attributed to the spatial heterogeneity of methanogens and methanotrophs across the island (Fig. 5), leading to an increase in methane production and a decrease in methane consumption from the island edge to the center. The methane sink at the island edge (Fig. 4) was mainly attributed to the strong oxidation by methanotrophs, which may consume methane to below equilibrium with the atmosphere. Methane emissions may not only rely on bacterial abundance but also bacterial activity (Schwarz et al., 2008). This deserves further studies using other molecular biology techniques, such as DNA/RNA-based stable isotope probing (Dumont and Murrell, 2005).

Groundwater DOC and sediment OC at the island edge, which are carbon sources for methane emission, were higher than that at the island center (Fig. 2b,c), suggesting that both sediment heterogeneity and dilution effects of hyporheic exchange had limited contribution to the spatial pattern of methane emissions in the island. Hyporheic exchange effectively shifted oxygen gradients across the island, resulting in substantial mitigation of potential methane emissions. In this study, only 0.2 % of the island area maintained a high methane flux (9.6–11.2 mg $h^{-1}m^{-2}$) (Fig. 4b), suggesting that methane emissions across the small island were attenuated, but only in the drawdown area where hyporheic exchange occurred.

**4.3 Implications**

Greenhouse gas emissions significantly detract from the green credentials of hydropower, and have thus received considerable research attention (Giles, 2006; Hu and Cheng, 2013). Previous studies have revealed that damming causes significant retention of carbon and creates deep, anoxic sediment strata, fueling methanogenesis and net water-air methane flux (Maeck et al., 2013). This study demonstrated that methane emissions at the most area of the sediment-deposited island were generally lower than the adjacent reservoir under reservoir operation (Fig. 4a), but higher than the drawdown area at other reservoir bank, such as Three Gorge Reservoir (Chen et al., 2011). This was mainly due to the deep sediment strata (about 60 m in depth) in the island. Given the widely distributed sediment-deposited islands in reservoirs, it should be of concern in future estimations of greenhouse gas emissions from dammed rivers. Prospective studies should assess the quantitative relationship between methane emissions from the drawdown area and hydropower operation scenarios.

Until now, few studies have concentrated on organic carbon mineralization in the drawdown area in reservoirs, with most focusing on the process of denitrification (Zarnetske et al., 2011b). Carbon emissions in the drawdown area are poorly understood, especially in regulated and dammed rivers. This study fills the knowledge gap and adds to our understanding of the ecological impacts of hydropower exploitation. Under reservoir operation, variable oxygen conditions and methane production may also affect the mercury cycle in the drawdown area and thereby the release of methylmercury (a bioaccumulative environmental toxicant) to the river (Marvin-DiPasquale et al., 2009), a subject deserving of further study.

**5 Conclusions**

In dammed rivers, sediment deposited islands are widely distributed in sidebays and are potential hotspots of methane emission to the atmosphere. In this study, high methane fluxes were only observed at the island center, while a ring-like zone of low methane emission or even sink was found in the drawdown area around the island edge. We attribute this spatial heterogeneity of methane emissions to hyporheic exchange between the reservoir and island. Under reservoir operation, frequent water level fluctuations drove hyporheic exchange, creating oxygen gradients along the hyporheic flowpath. These oxygen gradients affected the microbial communities associated with methane production and consumption, producing the spatial heterogeneity in methane emissions across sediment-deposited islands. This study will help us to evaluate the global warming effects of hydropower systems.

**Data availability**

The data presented here can be obtained upon request to Wenqing Shi (wqshi@nhri.cn).

**Author contribution**

Qiuwen Chen designed the research; Wenqing Shi, Yuyu Ji and Yuchen Chen performed the research; Cheng Chen and Juhua Yu contributed new reagents/analytic tools; Qiuwen Chen and Wenqing Shi analyzed the data; Jianyun Zhang and Bryce R. Van Dam contributed significant discussions and inputs; Wenqing Shi and Qiuwen Chen wrote the paper with input from all authors.

**Competing interests**

The authors declare that they have no conflict of interest.

**Acknowledgements**

Funding for this study was provided by the National Nature Science Foundation of

China (No. 91547206, 51425902, 51709181 and 51709182).

[Figure]

**Fig. 1** Map of the studied island in Manwan reservoir, Lancang-Mekong River.

[Figure]

**Fig. 2** Physicochemical properties of the island and reservoir, including water (a) DO, (b) DOC, and (c) sediment OC. DO = dissolved oxygen, DOC = dissolved oxygen carbon, OC = organic carbon, R = reservoir, W = monitoring wells. Error bars indicate standard deviations.

[Figure]

**Fig. 3** Vertical water level fluctuation (a) and horizontal hyporheic flow rate (b). R =

reservoir, W = monitoring wells. Positive fluxes indicate net flow from the reservoir to island, whereas negative values indicate net flow from the island to reservoir.

[Figure]

**Fig. 4** Methane emissions from the island and reservoir. (a) Spatial pattern of methane emissions; (b) Percentage of the island area emitting methane at a certain flux. Methane fluxes were interpolated separately for the island and reservoir.

[Figure]

**Fig. 5** Abundances of sediment methanogens and methanotrophs in the island. (a)

Spatial patterns of methanogen and methanotroph abundances across the island; (b) The ratio of methanogen to methanotroph abundance at each site. W= monitoring wells.

Error bars indicate standard deviations.

---

## Editor Comment (EC1) · J.-H. Park (Editor) · 23 Nov 2018

Dear Authors,

You have upload the first AC file here. Please correct it.

Sincerely,

Ji-Hyung Park, Associate Editor, Biogeosciences

---

## Author Comment (AC3) · 24 Nov 2018

**Referee # 2**

The manuscript presents an interesting strategy to estimate the spatial contribution of CH4 emissions from a sediment-deposited island in the Lancang-Mekong reservoir. While most of the studies in reservoirs are focused in estimate CH4 emissions in water sites, this study calls the attention to consider the sediment-island CH4 dynamics during reservoir operations (water level fluctuations). I think the study is relevant as strategy in such ecosystems (sediment-deposited island), but there are major clarifications that should be addressed before going a step forward in the acceptance.

**Microbial analysis issues.**

- 1. Due to the methodology of sampling (no with a core system) and different qPCR programs (I understand but you need explain it); methanotrophy and methanogenesis statements are overstated in the whole manuscript. I would recommend focusing more into the water level fluctuation to explain the CH4 dynamics obtained. Additionally, Figure 5 contains an artifact in the axis to pronounce your statements in the manuscript, please use the same axis level to avoid confusion, even a statistical analysis comparing each site (methanogens vs methanotrophs) and whole samples may help to approve such patterns.
  - Many thanks for the valuable suggestions that have enabled us to improve the manuscript.
  - In this study, frequent reservoir operation induced lateral hyporheic exchange and created oxygen gradients along the hyporheic flow path across the island (Fig. 1), regulating microbes for CH4 emissions. We analyzed the horizontal heterogeneity of microbial abundance here to clarify spatial patterns of CH4 emission. For the

vertical heterogeneity of bacterial community structure, we can design and conduct further studies using a core system according to the referee's suggestions.

- The qPCR program mainly depends on the sequence and length of functional genes and primers used, and different qPCR programs for archaeal 16S rDNA and *pomA* were applied in this study. This information has been updated. Please see line 176-181 on page 9 in the revised manuscript.
- In this study, we found that microbial processes regulated by water level fluctuation were responsible for CH4 emission patterns, had presented more statements on water level fluctuation than microbial processes (See the Discussion Section). Nevertheless, we have further simplified the statements about methanotrophy and methangenesis according to the referee's suggestions. Please see line 272-275 on page 14 in the revised manuscript.
- Methanogens and methanotrophs are two different microbial species, and their abundances often have different orders of magnitude in nature. For the comparison between methanogens and methanotrophs, it might be more meaningful to compare their relative values among sampling sites (Fig. 2 below). The Fig.2 below has been combined into Fig.5 and this information has been accordingly updated. Please see line 234-235 on page 12 in the revised manuscript.

Fig. 1 The horizontal heterogeneity in the interior of island under reservoir operation

Fig. 2 The ratio of methanogen to methanotroph abundance at different sampling sites

CH4 measurements.

- 2. While, the title of the manuscript is about CH4 emission. You have only one figure (Figure 4), and a small result description from the same figure (section 3.3) about CH4 emission patterns. Moreover, Figure 4 is over interpreted by the extrapolation, which is valid, but you should explain the range of uncertain.
  - The objective of this study is to explore spatial patterns of CH4 emissions from the sediment-deposited island and the underlying mechanisms. The data were presented in the manuscript as follows: Under reservoir operation, frequent water level fluctuations enhanced hyporheic exchange (Fig. 3) and created oxygen gradients (Fig. 2) along the hyporheic flow path, regulating microbial processes (Fig. 5) for CH4 emission patterns (Fig. 4). We have revised the title to "Methane emissions from a sediment-deposited island in a Lancang-Mekong reservoir: Spatial heterogeneity and mechanisms" to make it more appropriate.
  - The range of uncertain was 0.05 mg/h m2. This information has been updated, and please see line 160-161 on page 8 in the revised manuscript.
- 3. I have seen CH4 consumption in flux measurements in soils, but in water determinations are very strange. If you are confident of your measurements, I encourage you to include the data obtained, and explore relations with other parameters measured in the study.

- Thanks for the comments. There might be some misunderstanding here. It is CH4 production in flux measurements in water determinations. All the original data about CH4 fluxes at each sampling site (Fig. 3) are presented in Table 1 below.
- In this study, the oxygen gradients induced by reservoir operation were responsible for the spatial patterns of oxygen-sensitive CH4 emissions. Some other parameters, such as temperature, EC and pH, are indeed irrelevant and have been removed, and please see line 195-201 on page 10 in the revised manuscript.

Fig. 3 The map of sampling sites in this study

Table 1 The data about CH4 fluxes at each sampling site

| Sites |       | CH 4 concentration in the chamber (×10 -6 mol/L) |        |        |        |        |                           | Flux                                      | Flux                           | Average flux                   |
|-------|-------|------------------------------------------------------------------------|--------|--------|--------|--------|---------------------------|-------------------------------------------|--------------------------------|--------------------------------|
| Sites |       | 0 min                                                                  | 10 min | 20 min | 30 min | 40 min | (×10 -6 mol/h) | (×10 -6 mol/h m 2 ) | ( mg/h m 2 ) | ( mg/h m 2 ) |
| R1    | Rep-1 | 0.452                                                                  | 0.773  | 1.174  | 1.375  | 1.656  | 1.806                     | 57.5                                      | 0.92                           |                                |
|       | Rep-2 | 0.641                                                                  | 0.921  | 1.280  | 1.680  | 2.160  | 2.279                     | 72.6                                      | 1.16                           | 1.07                           |
|       | Rep-3 | 0.541                                                                  | 1.106  | 1.272  | 1.637  | 2.103  | 2.193                     | 69.8                                      | 1.12                           |                                |
| R2    | Rep-1 | 0.797                                                                  | 0.887  | 1.232  | 1.778  | 2.081  | 2.075                     | 66.1                                      | 1.06                           |                                |
|       | Rep-2 | 1.166                                                                  | 1.431  | 1.722  | 2.112  | 2.778  | 2.343                     | 74.6                                      | 1.19                           | 1.13                           |
|       | Rep-3 | 0.823                                                                  | 0.869  | 1.197  | 3.825  | 4.273  | NA                        | NA                                        | NA                             |                                |
| R3    | Rep-1 | 0.678                                                                  | 1.075  | 1.809  | 2.243  | 3.264  | 3.804                     | 121.1                                     | 1.94                           |                                |
|       | Rep-2 | 0.822                                                                  | 1.261  | 2.282  | 2.702  | 3.704  | 4.323                     | 137.7                                     | 2.20                           | 2.10                           |
|       | Rep-3 | 1.243                                                                  | 1.543  | 2.246  | 2.969  | 4.043  | 4.216                     | 134.3                                     | 2.15                           |                                |

| R4        | Rep-1 | 0.573 | 0.660  | 0.759  | 1.018  | 1.189  | 0.954  | 30.4  | 0.49  |       |
|-----------|-------|-------|--------|--------|--------|--------|--------|-------|-------|-------|
|           | Rep-2 | 0.390 | 0.553  | 0.665  | 0.797  | 0.829  | 0.673  | 21.4  | 0.34  | 0.39  |
|           | Rep-3 | 0.476 | 0.622  | 0.762  | 0.863  | 0.909  | 0.665  | 21.2  | 0.34  |       |
| R5        | Rep-1 | 1.867 | 2.871  | 4.802  | 6.432  | 8.739  | 10.383 | 330.7 | 5.29  |       |
|           | Rep-2 | 1.294 | 2.554  | 4.565  | 6.582  | 9.349  | 12.082 | 384.8 | 6.16  | 5.47  |
|           | Rep-3 | 1.188 | 2.165  | 4.289  | 4.913  | 7.935  | 9.745  | 310.4 | 4.97  |       |
|           | Rep-1 | 0.493 | 1.175  | 1.608  | 2.142  | 2.678  | 3.203  | 102.0 | 1.63  |       |
| R6        | Rep-2 | 0.881 | 1.411  | 1.881  | 2.751  | 3.561  | 4.021  | 128.1 | 2.05  | 1.84  |
|           | Rep-3 | 0.677 | 1.432  | 1.381  | 2.988  | 3.463  | NA     | NA    | NA    |       |
|           | Rep-1 | 0.388 | 0.802  | 0.923  | 1.224  | 1.482  | 1.566  | 49.9  | 0.80  |       |
| R7 | Rep-2 | 0.653 | 0.876  | 1.875  | 1.896  | 2.165  | NA     | NA    | NA    | 0.84  |
|           | Rep-3 | 0.794 | 0.930  | 1.420  | 1.609  | 1.902  | 1.737  | 55.3  | 0.89  |       |
|           | Rep-1 | 0.971 | 1.626  | 2.515  | 3.401  | 4.015  | 4.718  | 150.3 | 2.40  |       |
| R8        | Rep-2 | 0.879 | 0.997  | 1.377  | 1.995  | 2.469  | 2.508  | 79.9  | 1.28  | 1.84  |
|           | Rep-3 | 0.726 | 0.726  | 1.543  | 1.945  | 2.807  | 3.605  | 114.8 | 1.84  |       |
|           | Rep-1 | 0.463 | 0.434  | 0.358  | 0.295  | 0.200  | -0.399 | -12.7 | -0.20 | -0.21 |
| 1         | Rep-2 | 0.454 | 0.414  | 0.358  | 0.255  | 0.157  | -0.452 | -14.4 | -0.23 |       |
|           | Rep-3 | 0.386 | 0.314  | 0.258  | 0.214  | 0.139  | -0.356 | -11.3 | -0.18 |       |
| 2         | Rep-1 | 0.878 | 1.213  | 1.556  | 2.009  | 2.810  | 2.796  | 89.0  | 1.42  |       |
|           | Rep-2 | 0.784 | 0.970  | 1.432  | 1.945  | 2.442  | 2.575  | 82.0  | 1.31  | 1.26  |
|           | Rep-3 | 0.993 | 1.187  | 1.534  | 1.898  | 2.358  | 2.065  | 65.8  | 1.05  |       |
| 3         | Rep-1 | 1.068 | 1.483  | 2.425  | 3.567  | 4.738  | 5.654  | 180.1 | 2.88  |       |
|           | Rep-2 | 1.282 | 1.919  | 2.799  | 3.304  | 4.482  | 4.671  | 148.8 | 2.38  | 2.78  |
|           | Rep-3 | 1.482 | 1.949  | 2.876  | 3.967  | 5.519  | 6.055  | 192.8 | 3.09  |       |
| 4         | Rep-1 | 2.893 | 6.897  | 8.251  | 12.608 | 16.811 | 20.13  | 641.1 | 10.26 |       |
|           | Rep-2 | 3.892 | 7.996  | 10.451 | 15.906 | 17.211 | 20.73  | 660.2 | 10.56 | 10.41 |
|           | Rep-3 | 1.278 | 2.301  | 4.061  | 5.868  | 8.291  | NA     | NA    | NA    |       |
|           | Rep-1 | 0.786 | NA     | 2.784  | 4.324  | NA     | NA     | NA    | NA    |       |
| 5         | Rep-2 | 0.876 | 1.805  | 3.180  | 4.625  | 5.769  | 7.564  | 240.9 | 3.85  | 4.27  |
|           | Rep-3 | 1.069 | 2.075  | 3.208  | 5.141  | 7.203  | 9.201  | 293.0 | 4.69  |       |
|           | Rep-1 | 1.221 | 1.519  | 1.697  | 1.875  | 1.933  | 1.068  | 34.0  | 0.54  |       |
| 6         | Rep-2 | 0.782 | 0.819  | 0.999  | 1.153  | 1.338  | 0.868  | 27.6  | 0.44  | 0.49  |
|           | Rep-3 | 1.887 | 1.267  | 1.756  | 2.154  | 2.375  | NA     | NA    | NA    |       |
| -         | Rep-1 | 0.456 | 0.476  | 0.481  | 0.493  | 0.507  | 0.071  | 2.3   | 0.04  | 0.02  |
| 7         | Rep-2 | 0.479 | 0.489  | 0.499  | 0.505  | 0.515  | 0.05   | 1.0   | 0.03  | 0.05  |
|           | Rep-3 | 0.378 | 0.387  | 0.002  | 0.011  | 0.02   | 0.003  | 2.1   | 0.03  |       |
|           | Rep-1 | 0.399 | 0.408  | 0.417  | 0.422  | 0.428  | 0.045  | 1.4   | 0.02  | 0.02  |
| 0         | Rep-2 | 0.430 | 0.471  | 0.482  | 0.505  | 0.512  | 0.080  | 2.7   | 0.04  | 0.05  |
|           | Rep-3 | 0.400 | 0.497  | 0.505  | 0.517  | 0.519  | 0.049  | 1.0   | 0.02  |       |
| 0         | Rep-1 | 0.078 | 0.0708 | 0.531  | 0.528  | 0.521  | -0.003 | -0.2  | 0.00  | 0.01  |
| ,         | Rep-2 | 0.487 | 0.330  | 0.331  | 0.528  | 0.321  | -0.032 | -1.0  | -0.02 | -0.01 |
|           | Rep-1 | 0.407 | 0.703  | 0.40   | 0.583  | 0.472  | 0.023  | -0.7  | -0.01 |       |
| 10        | Rep-2 | 0.495 | 0.502  | 0.506  | 0.512  | 0.519  | 0.035  | 1.2   | 0.02  | 0.02  |
| 10        | Rep-2 | 0.785 | 0.701  | 0.500  | 0.802  | 0.517  | 0.035  | 1.1   | 0.02  | 0.02  |
|           | Kep-5 | 0.765 | 0.791  | 0.795  | 0.802  | 0.009  | 0.0555 | 1.1   | 0.02  |       |

|    | Rep-1 | 0.555 | 0.5545 | 0.5538 | 0.5535 | 0.5528 | -0.003 | -0.1  | 0.00  |       |
|----|-------|-------|--------|--------|--------|--------|--------|-------|-------|-------|
| 11 | Rep-2 | 0.453 | 0.454  | 0.454  | 0.455  | 0.456  | 0.004  | 0.1   | 0.00  | 0.00  |
|    | Rep-3 | 0.768 | 0.745  | 0.755  | 0.777  | 0.815  | NA     | NA    | NA    |       |
|    | Rep-1 | 0.787 | 1.199  | 1.443  | 2.108  | 2.588  | 2.707  | 86.2  | 1.38  |       |
| 12 | Rep-2 | 0.722 | 0.805  | 1.119  | 1.403  | 1.844  | 1.705  | 54.3  | 0.87  | 1.12  |
|    | Rep-3 | 0.477 | 0.674  | 0.846  | 0.876  | 1.532  | NA     | NA    | NA    |       |
|    | Rep-1 | 0.491 | 1.118  | 1.694  | 2.171  | 2.847  | 3.46   | 110.2 | 1.76  |       |
| 13 | Rep-2 | 0.493 | 1.015  | 1.290  | 1.965  | 2.390  | 2.85   | 90.8  | 1.45  | 1.61  |
|    | Rep-3 | NA    | 0.564  | NA     | 0.762  | 1.438  | NA     | NA    | NA    |       |
|    | Rep-1 | 0.393 | 0.444  | 0.477  | 0.521  | 0.532  | 0.021  | 0.7   | 0.01  |       |
| 14 | Rep-2 | 0.420 | 0.425  | 0.430  | NA     | 0.439  | 0.029  | 0.9   | 0.01  | 0.01  |
|    | Rep-3 | 0.422 | 0.427  | 0.432  | 0.435  | 0.440  | 0.0265 | 0.8   | 0.01  |       |
|    | Rep-1 | 0.754 | 0.733  | 0.695  | 0.677  | 0.644  | -0.166 | -5.3  | -0.08 |       |
| 15 | Rep-2 | 0.472 | 0.463  | 0.455  | 0.447  | 0.435  | -0.054 | -1.7  | -0.03 | -0.05 |
|    | Rep-3 | 0.511 | 0.493  | 0.475  | 0.457  | 0.447  | -0.098 | -3.1  | -0.05 |       |
|    | Rep-1 | 0.567 | 0.540  | 0.525  | 0.511  | 0.503  | -0.094 | -3.0  | -0.05 |       |
| 16 | Rep-2 | 0.853 | 0.840  | 0.835  | 0.821  | 0.809  | -0.064 | -2.0  | -0.03 | -0.04 |
|    | Rep-3 | 0.454 | 0.440  | 0.435  | 0.411  | 0.398  | -0.085 | -2.7  | -0.04 |       |
|    | Rep-1 | 0.263 | 0.564  | 0.672  | 0.675  | 0.785  | NA     | NA    | NA    |       |
| 17 | Rep-2 | 0.567 | 0.570  | 0.572  | 0.574  | 0.575  | 0.012  | 0.4   | 0.01  | 0.01  |
|    | Rep-3 | 0.368 | 0.370  | 0.375  | 0.377  | 0.381  | 0.02   | 0.6   | 0.01  |       |

- 4. Did you sample only once per time in each chamber? If you did it, you have large uncertain in your values, since there are a lot of risk to have leaks and lost sample before determining in the gas chromatograph. Which was the volume of gas pre-evacuated in each vial?
  - We are sorry for the unclear descriptions about gas flux analysis. Here, methane fluxes at each site were measured in triplicate by placing three individual chambers. This information has been updated, and please see line 157-158 on page 8 in the revised manuscript.
  - The volume of pre-evacuated vial is 12 ml. During gas collection, the vial was filled with 20 ml of gas sample for storage, and it is easy to draw 10 ml of gas sample from the vial using the syringe for analysis using gas chromatograph. The information about the vial size has been updated. Please see line 152 on page 8 in the revised manuscript.

General statements.

- 5. There are several statements that you are not able to support with your data, it would help you make a list of different parameters obtained and see how to relate them. Then, avoid such interpretation about redox changes, exudates, among others.
  - Thanks for the critical comments. We have deleted the statements that we are not able to support with our data, such as exudates and the strategy for the mitigation of methane emission hotspots in reservoir islands, and changed the description about redox changes to oxygen changes. Please see line 24 on page 2, line 67, 71 on page 4, line 76 on page 5, line 238 on page 12, line 260, 265 on page 13, line 285 on page 14, line 309 on page 15, and line 321, 322 on page 16 in the revised manuscript.
- 6. Sections 4.2 and 4.3 contain several discussions unnecessary (they are more introduction than discussion) about the relevance of the CH4 emission from aquatic ecosystems, how methane is produced biologically, the environment relevance of reservoirs, shelf mitigation, limitations using Darcy fluxes.
  - Many thanks for the valuable comments that have enabled us to improve the manuscript. We have deleted or removed the introduction-like discussions to Introduction Section (eg., the descriptions about hyporheic zone, the relevance of the CH4 emission from aquatic ecosystems, how methane is produced biologically), updated inappropriate discussions (e.g., shelf mitigation), and deleted unnecessary discussions (e.g., limitations using Darcy fluxes). Please see line 43-47 on page 3, line 81 on page 5, line 295-299 on page 15 and line 318, 323-325 on page 16 in the revised manuscript.

- 1 Methane emissions from a sediment-deposited island in a Lancang-Mekong
- 2 reservoir: Spatial heterogeneity and mechanisms
- 3 Wenqing Shi1,2, Qiuwen Chen1,2, Jianyun Zhang1,3, Cheng, Chen2, Yuchen Chen2, Yuyu
- 4 Ji2,4, Juhua Yu1,2, Bryce R. Van Dam5

[revised manuscript text omitted]
 outlet of the 145 chamber was open during the chamber deployment, and was left to stand for 20 min to 146 equilibrate with ambient pressure outside before sample collection. During gas 147 148 collection on the island, the chamber was carefully inserted 5 cm deep into the sediment, leaving 15 cm above the sediment surface. The outlet of the chamber was also open 149 during the chamber deployment, and was left to stand for 90 min to equilibrate before 150 151 sample collection. Gas samples (20 ml) were collected every 10 min over a 40-min period using a 25-ml polypropylene syringe and injected into a 12-ml pre-evacuated 152 Exetainer® vial (839 W, Labco, UK) for storage until analysis using a gas 153 154 chromatograph (7890B, Agilent Technologies, USA). Gas fluxes were calculated using linear regression based on the concentration changes of five samples over time. Linear 155 regression correlation coefficients of less than 0.95 were not accepted for further 156 calculations (Duchemin et al., 1999). Methane fluxes at each site were measured in 157 triplicate by placing three individual chambers. Simple spline interpolation was used to 158 interpolate the methane emissions from the sampling sites into space in the reservoir 159 and island separately (Immerzeel et al., 2009), and the range of the uncertain was 0.05 160  $mg h^{-1} m^{-2}$ . Methane emission areas at eight different categories were also calculated in 161

the island.

**163 **2.5 Microbial abundance analysis**

After being transported to the laboratory, the frozen sediment samples were stored 164 immediately at -80  $\,^{\circ}$ C for further molecular analysis. The sediment methanogens and 165 methanotrophs adjacent to each monitoring well across the island (ten sediment samples) 166 were quantified using qPCR. DNA extraction was undertaken using a FastDNA Power-167 Max Soil DNA Isolation Kit (MP Biomedical, USA) according to the manufacturer's 168 instructions. The qPCR assay was performed using primers targeting methanogenic 169 170 archaeal 16S rDNA (primer set, 1106F/1378R) and methanotrophic pmoA genes (primer set, A189F/M661R) (Watanabe et al., 2007; Ma and Lu, 2011). Gene copies 171 were amplified and quantified in a Bio-Rad cycler equipped with the iQ5 real-time 172 173 fluorescence detection system and software (version 2.0, Bio-Rad, USA). All reactions were completed in a total volume of 20 µL containing 10 µL SYBR® Premix Ex TagTM 174 (Toyobo, Japan), 0.5 mM of each primer, 0.8 μL of BSA (3 mg mL-1, Sigma, USA), 175 176 ddH2O, and template DNA. The qPCR program mainly depends on the sequence and length of functional genes and primers used, and different qPCR programs for archaeal 177 16S rDNA and *pomA* were applied in this study. The qPCR program for archaeal 16S 178 rDNA was as follows: 95  $^{\circ}$ C for 60 s, followed by 40 cycles of 95  $^{\circ}$ C for 25 s, 57  $^{\circ}$ C 179 for 30 s, and 72 °C for 60 s, and the qPCR program for *pomA* referred to: 95 °C for 60 180 s, followed by 40 cycles of 95 °C for 25 s, 53 °C for 30 s, and 72 °C for 60 s. A standard 181 curve was established by serial dilution  $(10^{-2}-10^{-8})$  of known concentration plasmid 182 DNA with the target fragment. All PCRs were run in triplicate on 96-well plates (Bio-183

184 Rad, USA) sealed with optical-quality sealing tape (Bio-Rad, USA). Three negative185 controls without the DNA template were included for each PCR run.

**186 **2.6 Data analysis**

One-way analysis of variance (ANOVA) was employed to test the statistical significance of differences between sampling sites. Post-hoc multiple comparisons of treatment means were performed using the Tukey's least significant difference procedure. All statistical calculations were performed using the SPSS (v22.0) statistical package for personal computers. The level of significance was P < 0.05 for all tests.

192

**193 **3 Results**

**3.1 Physicochemical characteristics**

As shown in Fig. 2, the island groundwater had lower DO but higher DOC, compared with that of the bulk reservoir water. Lateral gradients of groundwater DO and DOC were observed in the island. From the island edge to the center, DO and DOC decreased significantly from  $4.80 \pm 0.19$  to  $0.43 \pm 0.09$  mg L-1 and  $7.30 \pm 0.54$  to  $1.70 \pm 0.39$  mg L-1, respectively (P < 0.05) (Fig. 2a,b). In general, sediment OC was higher near the island edge, decreasing from  $6.37 \pm 0.69$  mg g-1 at the edge to  $2.42 \pm 0.60$  mg g-1 at the center of the island. Sediment OC in the reservoir was  $6.63 \pm 0.09$  mg g-1 (Fig. 2c).

**3.2 Water level fluctuation and hyporheic exchange**

203 The reservoir stage fluctuated frequently during the field survey, showing three distinct

peaks, with a maximum of 3.80 m in the first 37 h and gradual decline to below 1.30 m

in the next 60 h, yielding a maximum oscillation of 2.54 m. Similar oscillations were

| 206 | observed in the island water table, but were damped and lagged relatively to the                                             |
|-----|------------------------------------------------------------------------------------------------------------------------------|
| 207 | reservoir stage fluctuations (Fig. 3a). In W5, W7, and W10, the water levels reached                                         |
| 208 | 3.27, 3.41, and 3.33 m, then fell to 1.74, 2.09, and 2.01 m, for a maximum oscillation                                       |
| 209 | of 1.53, 1.33, and 1.32 m, respectively. Data from the automated water level recorders                                       |
| 210 | indicated that the water level responses in W5, W7, and W10 lagged the reservoir stage                                       |
| 211 | by 20, 25, and 30 min, respectively. Lateral hyporheic exchanges across the island bank                                      |
| 212 | were calculated according to the Darcy Law, showing that the flux was largest at the                                         |
| 213 | island edge and decreased from the edge to the center. The water exchange across the                                         |
| 214 | 0-10.5 m island edge zone was 1.2 and 4.7 times higher than those across the $10.5-20.5$                                     |
| 215 | m and 20.5–35.5 m zones, respectively. The flow rates at the reservoir-W5, W5-W7,                                            |
| 216 | and W7-W10 zones were relatively consistent at -0.55-1.35, -0.89-0.28, and -0.39-                                            |
| 217 | 0.17 m 2 d -1 (Fig. 3b), resulting in a water exchange volume of 2.61, 2.26, and 0.56 m 3 , |
| 218 | respectively, over the 115-h observation period.                                                                             |

**219 **3.3 Methane emissions**

High methane emission rates were observed at the island sites, with a maximum of 10.4 220  $mg h^{-1}m^{-2}$  at the center. However, a large ring-like low methane emission zone appeared 221 at the drawdown area around the island edge, where the methane flux was maintained 222 at -0.2–1.6 mg  $h^{-1}m^{-2}$  (Fig. 4a). The negative flux values also suggest the occurrence of 223 a methane sink at the island edge. The ring-like zone accounted for 89.1 % of the island 224 area, of which 9.1 % accounted for the methane sink zone (Fig. 4b). Compared with the 225 island, the methane flux from the adjacent reservoir was moderate at 0.4–5.5 mg h-1m- 226 2 (Fig. 4a). 227

**228 **3.4 Methanogen and methanotroph abundances**

Methanogens and methanotrophs were distributed non-uniformly across the island. In 229 230 general, methanogen counts were low at the island edge but high at the center, whereas methanotrophs were abundant at the island edge but scarce in the center. From the island 231 232 edge to the center, the methanogenic archaeal 16S rDNA gene increased from 0.12  $\times$  $10^5$  to 5.34  $\times 10^5$  copies g-1, and the methanotrophic *pmoA* gene decreased from 1.57  $\times$ 233  $10^6$  to  $0.64 \times 10^6$  copies g-1 (Fig. 5a). The ratio of methanogen to methanotroph 234 abundance increased from 0.01 at the island edge to 0.83 at the center (Fig. 5b). 235 236 **4** Discussion 237 4.1 Hyporheic exchange and oxygen gradients 238

In hydropower reservoirs, the release of water pulses is often employed to increase 239 power production and meet daily electricity peak demand (Bonalumi et al., 2012; 240 Toffolon et al., 2010). Such hydropeaking creates daily water level fluctuations in the 241 242 reservoir. In this study, frequent water level fluctuations were observed within the 115h observation period, with a maximum of 3.80 m (Fig. 3a). A hysteretic response 243 occurred in the island bank water table (Fig. 3a), driving water exchange between the 244 reservoir and island (Fig. 3b). The water exchange flux was largest close to the island 245 edge and decreased from the edge to the center, as water table fluctuations were 246 attenuated (Fig. 3a). 247

During a storage-release cycle, the island switched from water gaining to losing at daily or hourly scales, creating a ring-like drawdown area of enhanced hyporheic

exchange around the island. The drawdown area extended tens of meters into the island 250 bank (Fig. 3b). If the river system was unregulated, however, hydrodynamics within the 251 252 drawdown area would likely exhibit seasonal or annual patterns, or keep pace with snowmelt and rainstorm events, under a natural base flow-fed regime. In this case, the 253 254 drawdown area may be limited or altogether absent (Boano et al., 2008; Cardenas and Wilson, 2007). 255

[revised manuscript text omitted]

- hydroelectric reservoirs in China, Nature Climate Change, 3, 708, 2013.
- Hucks Sawyer, A., Bayani Cardenas, M., Bomar, A., and Mackey, M.: Impact of dam
- 388 operations on hyporheic exchange in the riparian zone of a regulated river, Hydrol
- 389 Process, 23, 2129-2137, 2009.
- Immerzeel, W., Rutten, M., and Droogers, P.: Spatial downscaling of TRMM
  precipitation using vegetative response on the Iberian Peninsula, Remote Sens
  Environ, 113, 362-370, 2009.
- 393 Klupfel, L., Piepenbrock, A., Kappler, A., and Sander, M.: Humic substances as fully
- regenerable electron acceptors in recurrently anoxic environments, Nat Geosci, 7,
- 395 195-200, 10.1038/NGEO2084, 2014.
- Li, J. P., Dong, S. K., Liu, S. L., Yang, Z. F., Peng, M. C., and Zhao, C.: Effects of
- cascading hydropower dams on the composition, biomass and biological integrity of
- 398 phytoplankton assemblages in the middle Lancang-Mekong River, Ecol Eng, 60,
- 399 316-324, 10.1016/j.ecoleng.2013.07.029, 2013.
- 400 Ma, K., and Lu, Y.: Regulation of microbial methane production and oxidation by
- 401 intermittent drainage in rice field soil, Fems Microbiol Ecol, 75, 446-456,
- 402 10.1111/j.1574-6941.2010.01018.x, 2011.
- 403 Maavara, T., Parsons, C. T., Ridenour, C., Stojanovic, S., Durr, H. H., Powley, H. R.,
- and Van Cappellen, P.: Global phosphorus retention by river damming, P Natl Acad
- 405 Sci USA, 112, 15603-15608, 10.1073/pnas.1511797112, 2015.
- 406 Maeck, A., DelSontro, T., McGinnis, D. F., Fischer, H., Flury, S., Schmidt, M., Fietzek,
- 407 P., and Lorke, A.: Sediment trapping by dams creates methane emission hot spots,

- 408 Environmental Science & Technology, 47, 8130-8137, 10.1021/es4003907, 2013.
- 409 Marvin-DiPasquale, M., Lutz, M. A., Brigham, M. E., Krabbenhoft, D. P., Aiken, G. R.,
- 410 Orem, W. H., and Hall, B. D.: Mercury cycling in stream ecosystems. 2. Benthic
- 411 methylmercury production and bed sediment-pore water partitioning, Environmental
- science & technology, 43, 2726-2732, 2009.
- 413 Naranjo, R. C., Niswonger, R. G., and Davis, C. J.: Mixing effects on nitrogen and
- 414 oxygen concentrations and the relationship to mean residence time in a hyporheic
- zone of a riffle-pool sequence, Water Resources Research, 51, 7202-7217,
- 416 10.1002/2014WR016593, 2015.
- Philip, J.: Approximate analysis of falling-head lined borehole permeameter, Water
  Resources Research, 29, 3763-3768, 1993.
- 419 Ran, L., Lu, X., Sun, H., Han, J., Li, R., and Zhang, J.: Spatial and seasonal variability
- of organic carbon transport in the Yellow River, China, Journal of Hydrology, 498,
  76-88, 2013.
- Rubol, S., Manzoni, S., Bellin, A., and Porporato, A.: Modeling soil moisture and
  oxygen effects on soil biogeochemical cycles including dissimilatory nitrate
- reduction to ammonium (DNRA), Advances in water resources, 62, 106-124, 2013.
- 425 Rubol, S., Freixa, A., Carles-Brangari, A., Fernàndez-Garcia, D., Roman í A., and
- 426 Sanchez-Vila, X.: Connecting bacterial colonization to physical and biochemical
- 427 changes in a sand box infiltration experiment, Journal of hydrology, 517, 317-327,
- 428 2014.
- 429 Schwarz, J. I. K., Eckert, W., Conrad, R.: Response of the methanogenic microbial

- 430 community of a profundal lake sediment (Lake Kinneret, Israel) to algal deposition.
- 431 Limnol. Oceanogr, 2008, 53(1), 113-121.
- 432 Seitzinger, S., Harrison, J., Dumont, E., Beusen, A. H., and Bouwman, A.: Sources and
- delivery of carbon, nitrogen, and phosphorus to the coastal zone: An overview of
- 434 Global Nutrient Export from Watersheds (NEWS) models and their application,
- 435 Global Biogeochem Cy, 19, 2005.
- 436 Smith, K. R., Desai, M. A., Rogers, J. V., and Houghton, R. A.: Joint CO2 and CH4
- 437 accountability for global warming, Proceedings of the National Academy of Sciences,
- 438 110, E2865-E2874, 2013.
- 439 Sobek, S., DelSontro, T., Wongfun, N., and Wehrli, B.: Extreme organic carbon burial
- 440 fuels intense methane bubbling in a temperate reservoir, Geophys Res Lett, 39, Artn
  441 L0140110.1029/2011gl050144, 2012.
- 442 Syvitski, J. P., V ör ösmarty, C. J., Kettner, A. J., and Green, P.: Impact of humans on the
- flux of terrestrial sediment to the global coastal ocean, science, 308, 376-380, 2005.
- 444 Thornton, K. W., Kimmel, B. L., and Payne, F. E.: Reservoir limnology: ecological
- 445 perspectives, John Wiley & Sons, 1990.
- 446 Toffolon, M., Siviglia, A., and Zolezzi, G.: Thermal wave dynamics in rivers affected
- 447 by hydropeaking, Water Resources Research, 46, Artn W08536
  448 10.1029/2009wr008234, 2010.
- Tonina, D., and Buffington, J. M.: Effects of stream discharge, alluvial depth and bar
  amplitude on hyporheic flow in pool-riffle channels, Water resources research, 47,
  2011.

| 452 | Wang, C., Liu, S. L., Zhao, Q. H., Deng, L., and Dong, S. K.: Spatial variation and        |
|-----|--------------------------------------------------------------------------------------------|
| 453 | contamination assessment of heavy metals in sediments in the Manwan Reservoir,             |
| 454 | Lancang River, Ecotox Environ Safe, 82, 32-39, 10.1016/j.ecoenv.2012.05.006, 2012          |
| 455 | Wang, X. W., Li, X. Z., Hu, Y. M., Lv, J. J., Sun, J., Li, Z. M., and Wu, Z. F.: Effect of |
| 456 | temperature and moisture on soil organic carbon mineralization of predominantly            |
| 457 | permafrost peatland in the Great Hing'an Mountains, Northeastern China, Journal of         |
| 458 | Environmental Sciences, 22, 1057-1066, 10.1016/S1001-0742(09)60217-5, 2010.                |
| 459 | Watanabe, T., Kimura, M., and Asakawa, S.: Dynamics of methanogenic archaeal               |
| 460 | communities based on rRNA analysis and their relation to methanogenic activity in          |
| 461 | Japanese paddy field soils, Soil Biol Biochem, 39, 2877-2887,                              |
| 462 | 10.1016/j.soilbio.2007.05.030, 2007.                                                       |
| 463 | Wilkinson, J., Maeck, A., Alshboul, Z., and Lorke, A.: Continuous seasonal river           |

- 464 ebullition measurements linked to sediment methane formation, Environmental465 science & technology, 49, 13121-13129, 2015.
- Wuebbles, D. J., and Hayhoe, K.: Atmospheric methane and global change, Earth-Sci
  Rev, 57, 177-210, 2002.
- 468 Yang, L., Lu, F., Wang, X., Duan, X., Song, W., Sun, B., Zhang, Q., and Zhou, Y.:
- 469 Spatial and seasonal variability of diffusive methane emissions from the Three
- 470 Gorges Reservoir, Journal of Geophysical Research: Biogeosciences, 118, 471-481,

471 2013.

- 472 Zarnetske, J. P., Haggerty, R., Wondzell, S. M., and Baker, M. A.: Labile dissolved
- 473 organic carbon supply limits hyporheic denitrification, Journal of Geophysical

- 474 Research: Biogeosciences, 116, 2011a.
- 475 Zarnetske, J. P., Haggerty, R., Wondzell, S. M., and Baker, M. A.: Labile dissolved
- 476 organic carbon supply limits hyporheic denitrification, Journal of Geophysical
- 477 Research, 116, 10.1029/2011jg001730, 2011b.
- 478

---

## Author Comment (AC4) · 17 Dec 2018

General comments: Thank you for providing your responses to the two referee reports. As the two reviewers mentioned, your manuscript reports on a very novel aspect of CH4 dynamics in a sediment-deposited island formed within a hydroelectric reservoir. However, both reviewers raised some critical issues about the rarely reported sink of CH4 at the water-sediment interface and potential flaws in measuring CH4 fluxes using chambers as described in the previous version. Given the importance of these issues and many additional uncertainties I myself found in reading the revised manuscript (please note that Biogeosciences require authors to provide the revised manuscript after this stage of editorial decision, not in writing your responses to reviewer comments), a major revision would be required to reconsider your manuscript for publication in

[Figure]

Biogeosciences. The revised manuscript may be resent to the reviewers to double check whether you have adequately addressed all the raised issues and minor corrections. I would like to ask you to make all the changes easily identifiable in a marked-up manuscript and a point-by-point reply to the reviewers' and my own comments to facilitate the second round of review. I would suggest you to specify the line numbers of the revised parts when you respond to the reviewer comments, and my comments and suggestions as follows:

- We are indeed grateful to the editor for the valuable comments that have enabled us to improve the manuscript. We seriously considered all the comments and made a thorough revision. The changes made were shown with a red color font in the marked-up manuscript and supplements. Detailed point-to-point responses are presented below. The marked-up manuscript and supplements are listed in the attachment.

Question 1: The CH4 sink along the island edge and accuracy of chamber measurements. Both reviewers were concerned about potential errors involved in your chamber measurements. Though you provided more detailed descriptions about the chamber system, there might be some lingering uncertainties about the repeated gas sampling in low-concentration chambers and potential gas leakage during storage of evacuation vials. I would suggest you to provide more details on sampling and gas analysis including QC measures to guarantee analytical accuracy so that the reviewers could assess the accuracy of CH4 measurements. You may also include the original CH4 measurements (as shown in your response to the reviewer 2) as a supplementary table. Lastly, you can also cite available literature information about very low to negative CH4 fluxes in reservoir sediment systems similar to yours. Your discussion in L256-267 is too general to explain specific conditions you encountered in your sediment island.

- Thanks for the editor's valuable comments. For gas sampling, static chambers have been widely used in the low-to-negative CH4 flux analyses (Veldkamp et al., 2013; Wang et al., 2013), and yielded a good linear regression based on CH4 concentration changes over time in this study (Table S1 in the supplements). For gas storage,

Exetainer® vials have been evaluated for storage of gas samples and they can limit sample loss to insignificant amounts for at least 3 months (Glatzel and Well, 2008; Van Dam et al., 2018). Nevertheless, we have presented more details on CH4 flux analysis and added the original CH4 data as a supplementary table according to the editor's suggestion. Please see line 148-151, 153-154, 160 on page 8, line 165, 166-168 on page 9, and the updated supplements.

- We have cited the literature information about very low-to-negative CH4 fluxes similar to this study. Please see line 151 on page 8, line 480-481 on page 23, and line 485-488 on page 24 in the revised manuscript.

- We have updated the discussion in Line 256-267 in the previous version to focus on specific conditions in the studied island. Please see line 277-283 on page 14 in the revised manuscript.

Wang, J. M., Murphy, J. G., Geddes, J. A., Winsborough, C. L., Basiliko, N., & Thomas, S. C. (2013). Methane fluxes measured by eddy covariance and static chamber techniques at a temperate forest in central Ontario, Canada. Biogeosciences, 10(6), 4371-4382.

Veldkamp, E., Koehler, B., & Corre, M. D. (2013). Indications of nitrogen-limited methane uptake in tropical forest soils. Biogeosciences, 10(8), 5367-5379.

Glatzel, S., & Well, R. (2008). Evaluation of septum-capped vials for storage of gas samples during air transport. Environmental Monitoring and Assessment, 136(1-3), 307-311.

Van Dam, B. R., Tobias, C., Holbach, A., Paerl, H. W., & Zhu, G. (2018). CO2 limited conditions favor cyanobacteria in a hypereutrophic lake: An empirical and theoretical stable isotope study. Limnology and Oceanography. https://doi.org/10.1002/lno.10798

Question 2: The quantitative importance of CH4 emissions from the anaerobic inner area of islands (spatial extent and seasonally) As you mentioned in the revised manuscript (L302-303) – "studies should assess the quantitative relationship between methane emissions from the drawdown area and hydropower operation scenarios", the information about hydropower operation and its effects on water levels and island flooding would be critical in assessing the quantitative significance of CH4 emissions from sediment-deposited islands. The revised manuscript still lacks this critical information. Please provide more details about the dam operation affecting reservoir and island water levels and dam discharge. If you want to argue for any enhancement or mitigation of CH4 emissions caused by fluctuating water levels and O2 availability, you also need to provide some estimates for the total area of islands affected by the processes you report in your manuscript and discuss their implications for the total reservoir CH4 emission.

- Thanks for the comments. The objective of this study is to explore spatial patterns of CH4 emissions from the sediment-deposited island and the underlying mechanisms. We found the spatial heterogeneity of methane emissions from the island and attributed it to the shift of oxygen availability in sediments under water level fluctuation induced by reservoir operation. The descriptions had been presented in the manuscript. Please see line 78-84 on page 5, line 284-300 on page 14-15, and line 324-335 on page 16. It is very interesting to further establish the quantitative relationship between methane emissions and water level fluctuation (or hydropower operation) and assess the mitigation of CH4 emissions from all the islands in these cascade reservoirs, however, this is a much more complex and difficult task that is beyond the aims of our current study. We can study it separately in the future with the help of China Huaneng Group Co., Ltd, who runs these cascade reservoirs. Here, the limited descriptions of "Prospective studies should assess the quantitative relationship between methane emissions from the drawdown area and hydropower operation scenarios" and "mitigation of CH4 emissions" are the implications based on the results of this study. To avoid confusion, we have deleted these descriptions.

Question 3: L56, L61: Please provide definitions of "sidebays" and "forebays".

- The forebay is an artificial pool of water in front of a larger body of water, which refers to the waters in front of the dam in reservoirs (Wikipedia). The sidebay is formed by abrupt variations in channel width (Jenkins et al., 1981), which refers to the corner of the reservoir here. Based on the definition of sidebays, we updated the number of sediment deposited islands in sidebays. Please see line 94 on page 5 in the revised manuscript and the updated supplements.

https://en.wikipedia.org/wiki/Forebay_(reservoir).

Jenkins, B. S. (1981). Effects of channel geometry on exchange processes in river side-bays. In Conference on Hydraulics in Civil Engineering 1981: Preprints of Papers (p. 156). Institution of Engineers, Australia.

Question 4: L64: What do you mean by "following hydropower production demands"?

- In reservoirs, the water level decreases during hydropower production, and increases with the inputs from inflows during the stagnant hydropower production. Hence, the water level frequently fluctuates following hydropower production demands. Here, we updated the description to make it clearer. Please see line 64-66 on page 4 in the revised manuscript.

Question 5: L67 & L71-73: It is difficult to understand why this happens - "This may lead to changes of oxygen conditions in the interior of the drawdown area". You need to provide more relevant background information about anaerobic conditions developing in sediment-deposited islands, not the very general description – "oxygen conditions in sediments may affect the microbial processes". I would suggest you to cite more relevant studies so that readers can understand better how anaerobic conditions in sediment systems similar to your study site affect methanogens and their competitive balance over methanotrophs.

- Many thanks for the valuable comments for improving the manuscript. As for the relevant background information about anaerobic conditions developing in deposited sediment, we had presented it in the paragraph ahead. Please see line 47-49 on page 3.

- As for the effects of oxygen conditions in sediments on the microbial processes, we have rewritten it and cited more relevant studies. Please see line 71-77 on page 3-4 in the revised manuscript.

Question 6: L79-81: Please be more specific in providing your study objective. I expected that you would examine how changing hydrologic conditions and oxygen-dependent metabolic processes affect CH4 production and consumption.

- Previous studies have mainly focused on CH4 emissions from dam forebays, while the understandings of CH4 emissions from sediments deposited in sidebays remain poor (Line 60-63, Page 4). The objective of this study is to explore spatial patterns of CH4 emissions from the sediment-deposited island in sidebays and the underlying mechanisms. The changing hydrologic conditions and oxygen-dependent metabolic processes were studied here to explore the mechanisms of CH4 emission spatial patterns across the sediment-deposited island. We have updated our study objective in line 83-84 on page 5 in the revised manuscript.

Question 7: L86-88: The basin information is quite different from other reported values. Please double check and provide more official (or common) values (e.g., Mekong River Commission reports or Milliman, J. D., Farnsworth, K. L.: River Discharge to the Coastal Ocean: A Global Synthesis, Cambridge University Press, Cambridge, UK, 2011.)

- Thanks for the editor's careful correction. The previous basin information was cited from Li et al. (2013). To be more official, we have updated this information from the Mekong River Commission according to the editor's suggestion. Please see line 88-92 on page 5 in the revised manuscript.

Li, J. P., Dong, S. K., Liu, S. L., Yang, Z. F., Peng, M. C., and Zhao, C.: Effects of cascading hydropower dams on the composition, biomass and biological integrity of phytoplankton assemblages in the middle Lancang-Mekong River, Ecol Eng, 60, 316-324, 10.1016/j.ecoleng.2013.07.029, 2013.

Question 8: L92-94: Please describe here key information about the dam.

- Thanks for the valuable comments. We have moved the key information about the Manwan dam from the supplements to the manuscript. Please see line 100-103 on page 6 in the revised manuscript.

Question 9: L95-100: This short site description lacks many important details about the studied island (How "typical" is it? How many those islands in the sidebay vs. forebay? Total areas of those islands and the study site's area?",), local climates (What is plateau monsoon climate? Rainfall and discharge regimes? Temperature?",), and reservoir operation?

- The forebay is in front of the dam in the reservoir, where sediment accumulation can not form islands but build up cohesive sediment layers at the bottom because of the deep water and the high flow rate near the hydropower stations within the dam. We had presented the description of "sediment accumulates in forebays and sidebay islands" in line 39-40 on page 3 in the revised manuscript. In the upstream section of the Lancang-Mekong River in China, there were about 33 sediment-deposited islands in the reservoir sidebay, of which 81.8% are located at the convex bank, and 18.2% at the concave bank. Hence, we selected the widely distributed type of island at the convex bank for investigation, which the "typical" island refers to in this study. To make it clearer, we have revised the description to "This study selected the widely distributed type of island at the convex bank for investigation, which is located in the Manwan Reservoir ". Please see line 93-98 on page 5-6 in the revised manuscript. The studied island has a surface area of $1.3 \times 10^4$ m2, and the total area of the 33 sediment-deposited islands was estimated to about $4.3 \times 10^5$ m2. The information has been also updated in line 94 on page 5, line 99 on page 6 in the revised manuscript.

- The subtropical plateau monsoon climate refers to that the air temperature features no distinct seasons. This sentence has been revised to "Manwan has a subtropical plateau monsoon climate, and the temperature features no distinct seasons". Please see line 103-104 on page 6 in the revised manuscript.

- The "reservoir operation" means "water level fluctuation induced by reservoir operation", which has been corrected to make it clearer. Please see line 105 on page 6 in the revised manuscript.

Question 10: L104: Your site map needs to distinguish clearly the locations and labels of groundwater wells from those for sediment sampling. It is now very confusing. Please think about showing some key site labels on the site map (Fig. 1), and providing more details in the supplementary information, like Fig. S3 or Fig. 3 in your response to the second reviewer comments.

- We thank for the editor's comments. The groundwater wells/sediment sampling sites and gas sampling sites were indicated in red and yellow dots, respectively, in Fig. 1b, and the labels were showed in Fig. 1a. To make it clearer, we have modified Fig. 1, and provided more details in the supplementary information. Please see Fig. 1 and the updated supplements.

Question 11: L122: Please describe how you kept the collected samples to prevent sample exposure to air.

- The samples were kept in the plastic ziplock bag to prevent exposure to air. This information has been updated, and please see line 130-131 on page 7 in the revised manuscript.

Question 12: L130: I wondered how you could measure DO in situ using the 6600 sonde. Was it possible to put in in the well with 9 cm diameter?

- We are very sorry for the wrong description here. The YSI 6600 was applied to the DO measurements in the reservoir, and another portable DO meter was used to measure the DO at each well by directly putting the probe to the well. This mistake has been corrected. Please see line 136-139 on page 7 in the revised manuscript.

Question 13: L160: What do you mean by "the range of the uncertain"?

- In this study, we used simple spline interpolation to interpolate the methane emissions from the island and adjacent reservoir based on several measured values. This method is valid, but there exist the range of uncertain, which is similar to error margin.

Question 14: L178: pomA?

- The pmoA gene is the most commonly used functional markers of methanotrophs, which encodes a subunit of the particulate methane monooxygenase (Nauer et al., 2012; Sheng et al., 2016).

Nauer, P. A., Dam, B., Liesack, W., Zeyer, J., & Schroth, M. H. (2012). Activity and diversity of methane-oxidizing bacteria in glacier forefields on siliceous and calcareous bedrock. Biogeosciences, 9(6), 2259-2274.

Sheng, R., Chen, A., Zhang, M., Whiteley, A. S., Kumaresan, D., & Wei, W. (2016). Transcriptional activities of methanogens and methanotrophs vary with methane emission flux in rice soils under chronic nutrient constraints of phosphorus and potassium. Biogeosciences, 13(23), 6507-6518.

Question 15: L198: Please also provide % DO values to allow for a better assessment of O2 availability.

- Thanks for the valuable suggestion. We have changed DO concentrations to % DO values. Please see line 211 on page 11 and Figure 2 in the revised manuscript.

Question 16: L220: At which "island sites"?

- Here, it means that high CH4 emissions were observed at the sites on the island other than the reservoir. To avoid confusion, we have revised this sentence to "High methane emission rates were observed at the island center, with a maximum of 10.4

mg h-1m-2." Please see line 233-234 on page 11 in the revised manuscript.

Question 17: L220-227: Please describe the proportions represented by the edge sink, the low-emission ring, and the core source in the total island emission.

- The total island emission contained the emission from the low emission ring and the core source, which was estimated to 8.3 and 5.4 g h-1, accounting for 60.6% and 39.4%, respectively. This information has been updated in line 239-240 on page 12 in the revised manuscript. However, we are not sure how to get the proportions represented by the edge sink in the total island emission.

Question 18: L243 (and L206-207): Please describe how "hysteretic response" occurred. Delays in the response of islands water levels to rising reservoir levels appear only slight during the initial rising phase, and almost negligible on the descending limb of the water level.

- The hysteretic response occurred in presence of head gradients and sediment resistance (Gerecht et al., 2011). The information has been updated in the revised manuscript. Please see line 258-259 on page 13.

- Since the water permeability was high for sands, the hysteretic response is slight in the sand-deposited island. Data from the automated water level recorders indicated that the water level responses in the island lagged the reservoir stage by 20–30 min (line 222-224 on page 11). The delays on the descending limb of the water level become negligible due to the water conservation of sediments in the island.

Gerecht, K. E., Cardenas, M. B., Guswa, A. J., Sawyer, A. H., Nowinski, J. D., & Swanson, T. E. (2011). Dynamics of hyporheic flow and heat transport across a bed‐to‐bank continuum in a large regulated river. Water Resources Research, 47(3).

Question 19: L297: Under which reservoir operation? Do you mean under the conditions of fluctuating water levels?

- Yes, it refers to 'under the conditions of fluctuating water levels'. We have checked the similar descriptions throughout the manuscript and revised them to "under water level fluctuation induced by reservoir operation" to make them clearer. Please see line 105 on page 6, line 308-309 on page 15 and line 318-319 on page 16 in the revised manuscript.

Question 20: L298: Please provide the range reported for other reservoir banks.

- The range is -0.08–0.66 mg h-1m-2. This information has been updated in line 309-310 on page 15 in the revised manuscript.

Question 21: L301: Quantitatively the magnitude of "concern"? This statement is contradictory to what you suggested in the previous section (L286-289).

- Thanks for the careful corrections. We have deleted this inappropriate description here.

Please also note the supplement to this comment:
https://www.biogeosciences-discuss.net/bg-2018-380/bg-2018-380-AC4-supplement.pdf

**Supplement:**

**Methane emissions from a sediment-deposited island in a Lancang-Mekong**

**reservoir**: **Spatial heterogeneity and mechanisms**

[revised manuscript text omitted]

In reservoirs, the water level decreases during hydropower production, and increases with the inputs from inflows during the stagnant hydropower production. As a result, the water level frequently fluctuates following hydropower production demands, which enhances hyporheic exchange by driving water flow in and out of the drawdown area (Tonina and Buffington, 2011; Hucks Sawyer et al., 2009). This may lead to changes of oxygen conditions in the interior of the drawdown area. Zarnetske found a redox gradient along the hyporheic flow paths in a third-order stream in the Willamette River basin, USA (Zarnetske et al., 2011a). Methane from sediments is mainly produced by methanogens, and is consumed by methanotrophs (Borrel et al., 2011). Methanogens and methanotrophs belong to anaerobic and aerobic microbes, respectively (Nazaries et al., 2013). The oxygen conditions in sediments have been found to manipulate these microbial processes and methane emission (Chamberlain et al., 2016; Shi et al., 2018). Hence, we suppose the changes of oxygen conditions in sediments may alter methane emission scheme on sediment-deposited islands in reservoir sidebays.

In this study, methane emissions from a sediment-deposited island were investigated in the sidebay of Manwan Reservoir, Lancang-Mekong River. Monitoring wells were established to probe hyporheic exchange and oxygen gradients across the island. Methanogen and methanotroph abundances in the sediment were analyzed using quantitative polymerase chain reaction (qPCR) to reveal the associated molecular mechanism. The objective of this study was to explore methane emission patterns from sediment-deposited zones in the reservoir sidebay and the underlying mechanisms.

**2    Methods**

**2.1 Study area**

The Lancang-Mekong River is a trans-boundary river in Southeast Asia and the tenth-largest river in the world, which originates from the Tibetan Plateau in China, continues into Myanmar, Lao PDR, Thailand, Cambodia and Viet Nam, and discharges into the South China Sea. It has a length of approximately 4909 km, a total watershed area of 795,000 $km^2$ (Mekong River Commission, 2018). Due to rich hydropower resources, cascade dams have been built on the mainstream of the Lancang-Mekong River. After impoundment, about 33 sediment-deposited islands of $4.3 \times 10^5$ $m^2$ formed in the reservoir sidebay in the upstream section of the Lancang-Mekong River in China, of which 81.8% are located at the convex bank, and 18.2% at the concave bank (Fig. S1). This study selected the widely distributed type of island at the convex bank for investigation, which is located in the Manwan Reservoir (24°43′44″ N, 100°23′5″ E). The studied island has an oval shape with a surface area of $1.3 \times 10^4$ m$^2$ (Fig. 1). Manwan dam is the first-built one (completed in 1993) in the upstream section of the Lancang-Mekong River in China, with a height of 132 m. The Manwan Reservoir has a normal water level of 994 m, a total storage capacity of $5.0 \times 10^8$ m$^3$, an installed capacity of $1.5 \times 10^6$ kW and hydrological residence time of 0.78 a. Manwan has a subtropical plateau monsoon climate, and the temperature features no distinct seasons. Under water level fluctuation induced by reservoir operation, the island bank is frequently flooded (Fig. S2).

**2.2 Monitoring wells**

Ten monitoring wells were installed in the island bank at 0.5 (W1), 1.5 (W2), 3.5 (W3), 6.5 (W4), 10.5 (W5), 15.5 (W6) 20.5 (W7), 25.5 (W8), 30.5 (W9), and 35.5 m (W10) from the waterline, respectively (Fig. S2). The wells were 90-mm diameter perforated polyvinylchloride pipes, reaching a depth of 2.0 m below the ground surface. To prevent flooding, the wells were extended 2.0 m aboveground. Due to hydropower production, the reservoir runs in a pseudo-periodic hydrological regime with cyclic water level fluctuations. Here, we monitored a complete cycle of water level fluctuation within 115 h. Water levels were measured every 10 min from 11 to 16 September 2016 using automated water level recorders (U2000101, OneSetHoBo, USA), which were mounted at the bottom of W5, W7, W10, and the reservoir (Fig. S3).

Instantaneous lateral fluid fluxes ($q$) across the island bank per unit length were calculated following the Darcy Eq. (1) (Gerecht et al., 2011; Hucks Sawyer et al., 2009):

$$q(t) = -Kb \cdot \left[ \frac{\vartheta h(x,t)}{\vartheta x} \right] \tag{1}$$

where $Kb$ is sediment transmissivity, m d$^{-1}$; $h$ is hydraulic head, m; $x$ is distance, m; and

$t$ is time, d. A positive $q$ value indicates flow from the reservoir to the island. The island

$Kb$ was 0.99 m d$^{-1}$, which was measured according to Philip (1993).

**2.3 Sampling and physicochemical analysis**

After water level receded at the monitoring time of 100 h, groundwater (100 ml) was carefully sampled in triplicate from each monitoring well with a portable peristaltic pump (SC-1/253Yx, Chongqing Jieheng Peristaltic Pump Co., Ltd., China), and then filtered *in situ* using portable syringe filters for water DOC analysis. Sediment (5 g)

was synchronously collected in triplicate from 10 cm below the surface adjacent to each well (Fig. 1b) using a hand shovel, and then quickly homogenized before the storage in the plastic ziplock bag for the analyses of sediment OC and microbe. At a reservoir site adjacent to W1, water and surface sediment samples were also collected in triplicate using a stainless-steel bucket and an Ekman grab sampler, respectively. The collected water and sediment samples were kept frozen in an ice box (-5 ℃–-10 ℃) and transported to the laboratory for analysis within three days.

Dissolved oxygen (DO) in the reservoir was measured using a multi-sensor probe (YSI 6600, Yellow Springs Instruments, USA), and the DO at each well was measured

*in situ* using a DO meter (JPB-607A, Shanghai INESA Scientific Instrument Co. Ltd.,

China) by directly placing the probe in the well. Analysis of dissolved organic carbon (DOC) in the water was conducted on filtered samples (Whatman GF/F, UK) using a total organic carbon analyzer (Liqui TOC II, Elementar Inc., Germany). Sediment OC

was determined using a vario MACRO cube elementar (Elementar Inc., Germany).

Fresh sediment was freeze-dried and ground before analysis. Approximately 30 mg of each sample was weighed in a tin cup and acidified with two drops of 8 % $H_3PO_4$ to remove inorganic carbonates before OC analysis.

**2.4 Methane flux analysis**

Methane fluxes from the reservoir (eight sampling sites) and island (seventeen sampling sites) were analyzed using the static chamber method (Duchemin et al., 1999). The static chamber method has been widely used in the analyses of methane fluxes from air-soil/water interfaces, including the low-to-negative methane flux analyses (Veldkamp et al., 2013; Wang et al., 2013). The sampling sites are shown in Fig. 1b and Fig. S4. The plexiglass chamber consisted of a 6.28-L cylinder (20 cm in diameter,

20 cm in height) and a removable Styrofoam collar, and was covered with Mylar paper to prevent temperature rise inside the chamber when exposed to the sun. During gas collection in the reservoir, the chamber was fitted with the Styrofoam collar, which maintained the upper closed portion of the chamber about 10 cm above the water surface (Fig. S5). The outlet of the chamber was open during the chamber deployment, and was left to stand for 20 min to equilibrate with ambient pressure outside before sample collection. During gas collection on the island, the chamber was carefully inserted 5 cm deep into the sediment according to (Smith et al., 2000), leaving 15 cm above the sediment surface. The outlet of the chamber was also open during the chamber deployment, and was left to stand for 90 min to equilibrate before sample collection. Gas samples (20 ml) were collected every 10 min over a 40-min period using a 25-ml polypropylene syringe and injected into a 12-ml pre-evacuated Exetainer® vial (839 W, Labco, UK) with double wadded septa caps for storage until analysis using a gas chromatograph (7890B, Agilent Technologies, USA). Exetainer® vials have been evaluated for storage of gas samples and they can limit sample loss to insignificant amounts for at least 3 months (Glatzel and Well, 2008; Van Dam et al., 2018). Gas fluxes were calculated using linear regression based on the concentration changes of five samples over time. Linear regression correlation coefficients of less than 0.95 were not accepted for further calculations (Duchemin et al., 1999). Methane fluxes at each site were measured in triplicate by placing three individual chambers. Simple spline interpolation was used to interpolate the methane emissions from the sampling sites into space in the reservoir and island separately (Immerzeel et al., 2009), and the range of the uncertain was 0.05 mg h$^{-1}$ m$^{-2}$. Methane emission areas at eight different categories were also calculated in the island.

**2.5 Microbial abundance analysis**

After being transported to the laboratory, the frozen sediment samples were stored immediately at -80 ℃ for further molecular analysis. The sediment methanogens and methanotrophs adjacent to each monitoring well across the island (ten sediment samples)

were quantified using qPCR. DNA extraction was undertaken using a FastDNA Power-

Max Soil DNA Isolation Kit (MP Biomedical, USA) according to the manufacturer's instructions. The qPCR assay was performed using primers targeting methanogenic archaeal 16S rDNA (primer set, 1106F/1378R) and methanotrophic *pmoA* genes (primer set, A189F/M661R) (Watanabe et al., 2007; Ma and Lu, 2011). Gene copies were amplified and quantified in a Bio-Rad cycler equipped with the iQ5 real-time fluorescence detection system and software (version 2.0, Bio-Rad, USA). All reactions were completed in a total volume of 20 μL containing 10 μL SYBR® *Premix Ex Taq$^{TM}$*

(Toyobo, Japan), 0.5 mM of each primer, 0.8 μL of BSA (3 mg mL$^{-1}$, Sigma, USA), ddH$_2$O, and template DNA. The qPCR program mainly depends on the sequence and length of functional genes and primers used, and different qPCR programs for archaeal

16S rDNA and *pomA* were applied in this study. The qPCR program for archaeal 16S

rDNA was as follows: 95 ℃ for 60 s, followed by 40 cycles of 95 ℃ for 25 s, 57 ℃

for 30 s, and 72 ℃ for 60 s, and the qPCR program for *pomA* referred to: 95 ℃ for 60

s, followed by 40 cycles of 95 ℃ for 25 s, 53 ℃ for 30 s, and 72 ℃ for 60 s. A standard curve was established by serial dilution ($10^{-2}$–$10^{-8}$) of known concentration plasmid

DNA with the target fragment. All PCRs were run in triplicate on 96-well plates (Bio-

Rad, USA) sealed with optical-quality sealing tape (Bio-Rad, USA). Three negative controls without the DNA template were included for each PCR run.

**2.6 Data analysis**

One-way analysis of variance (ANOVA) was employed to test the statistical significance of differences between sampling sites. Post-hoc multiple comparisons of treatment means were performed using the Tukey's least significant difference procedure. All statistical calculations were performed using the SPSS (v22.0) statistical package for personal computers. The level of significance was $P < 0.05$ for all tests.

**3 Results**

**3.1 Physicochemical characteristics**

As shown in Fig. 2, the island groundwater had lower DO but higher DOC, compared with that of the bulk reservoir water. Lateral gradients of groundwater DO and DOC

were observed in the island. From the island edge to the center, DO and DOC decreased significantly from 50.7 $\pm$ 2.0 to 4.5 $\pm$ 0.9% and 7.30 $\pm$ 0.54 to 1.70 $\pm$ 0.39 mg L$^{-1}$, respectively ($P < 0.05$) (Fig. 2a,b). In general, sediment OC was higher near the island edge, decreasing from 6.37 $\pm$ 0.69 mg g$^{-1}$ at the edge to 2.42 $\pm$ 0.60 mg g$^{-1}$ at the center of the island. Sediment OC in the reservoir was 6.63 $\pm$ 0.09 mg g$^{-1}$ (Fig. 2c).

**3.2 Water level fluctuation and hyporheic exchange**

The reservoir stage fluctuated frequently during the field survey, showing three distinct peaks, with a maximum of 3.80 m in the first 37 h and gradual decline to below 1.30 m in the next 60 h, yielding a maximum oscillation of 2.54 m. Similar oscillations were observed in the island water table, but were damped and lagged relatively to the reservoir stage fluctuations (Fig. 3a). In W5, W7, and W10, the water levels reached

3.27, 3.41, and 3.33 m, then fell to 1.74, 2.09, and 2.01 m, for a maximum oscillation of 1.53, 1.33, and 1.32 m, respectively. Data from the automated water level recorders indicated that the water level responses in W5, W7, and W10 lagged the reservoir stage by 20, 25, and 30 min, respectively. Lateral hyporheic exchanges across the island bank were calculated according to the Darcy Law, showing that the flux was largest at the island edge and decreased from the edge to the center. The water exchange across the

0–10.5 m island edge zone was 1.2 and 4.7 times higher than those across the 10.5–20.5

m and 20.5–35.5 m zones, respectively. The flow rates at the reservoir-W5, W5-W7, and W7-W10 zones were relatively consistent at -0.55–1.35, -0.89–0.28, and -0.39–

0.17 $m^2$ $d^{-1}$ (Fig. 3b), resulting in a water exchange volume of 2.61, 2.26, and 0.56 $m^3$, respectively, over the 115-h observation period.

**3.3 Methane emissions**

High methane emission rates were observed at the island center, with a maximum of

10.4 mg $h^{-1}m^{-2}$. However, a ring-like zone of low-to-negative methane emission appeared at the drawdown area around the island edge, where the methane flux was maintained at -0.2–1.6 mg $h^{-1}m^{-2}$ (Fig. 4a). The negative flux values also suggest the occurrence of a methane sink at the island edge. The ring-like zone accounted for 89.1 %

of the island area, of which 9.1 % accounted for the methane sink zone (Fig. 4b).

Methane emissions were estimated to 8.3 and 5.4 g $h^{-1}$ in total from the island center and the ring-like zone, accounting for 60.6% and 39.4%, respectively. Compared with the island, the methane flux from the adjacent reservoir was moderate at 0.4–5.5 mg $h^{-}$

$^1m^{-2}$ (Fig. 4a).

**3.4 Methanogen and methanotroph abundances**

Methanogens and methanotrophs were distributed non-uniformly across the island. In general, methanogen counts were low at the island edge but high at the center, whereas methanotrophs were abundant at the island edge but scarce in the center. From the island edge to the center, the methanogenic archaeal 16S rDNA gene increased from 0.12 $\times$

$10^5$ to 5.34 $\times 10^5$ copies $g^{-1}$, and the methanotrophic *pmoA* gene decreased from 1.57 $\times$

$10^6$ to 0.64 $\times 10^6$ copies $g^{-1}$ (Fig. 5a). The ratio of methanogen to methanotroph abundance increased from 0.01 at the island edge to 0.83 at the center (Fig. 5b).

**4 Discussion**

**4.1 Hyporheic exchange and oxygen gradients**

In hydropower reservoirs, the release of water pulses is often employed to increase power production and meet daily electricity peak demand (Bonalumi et al., 2012;

Toffolon et al., 2010). Such hydropeaking creates daily water level fluctuations in the reservoir. In this study, frequent water level fluctuations were observed within the 115- h observation period, with a maximum of 3.80 m (Fig. 3a). In presence of head gradients and sediment resistance (Gerecht et al., 2011), a hysteretic response occurred in the island bank water table (Fig. 3a), driving water exchange between the reservoir and island (Fig. 3b). The water exchange flux was largest close to the island edge and decreased from the edge to the center, as water table fluctuations were attenuated (Fig.

3a).

During a storage-release cycle, the island switched from water gaining to losing at daily or hourly scales, creating a ring-like drawdown area of enhanced hyporheic exchange around the island. The drawdown area extended tens of meters into the island bank (Fig. 3b). If the river system was unregulated, however, hydrodynamics within the drawdown area would likely exhibit seasonal or annual patterns, or keep pace with snowmelt and rainstorm events, under a natural base flow-fed regime. In this case, the drawdown area may be limited or altogether absent (Boano et al., 2008; Cardenas and

Wilson, 2007).

Exchange across the sediment-water interface involves mixing of surface water and groundwater through hyporheic flow (Hester et al., 2013; Naranjo et al., 2015). In this study, when the reservoir water entered the hyporheic flow path, it was typically rich in oxygen (Fig. 2d). As oxygen was consumed through aerobic respiration, other terminal electron acceptors were utilized (Klupfel et al., 2014), creating an oxygen gradient along the hyporheic flow path (Fig. 2d). This caused the spatial heterogeneity of the abundances of oxygen-sensitive methanogens and methanotrophs across the island. The oxygen-rich environment at the island edge favored methanotroph growth and inhibited methanogen growth, while the oxygen-poor environment at the island center inhibited methanotroph growth and favored methanogen growth. As a result, we detected low methanogen abundance at the island edge, but high abundance at the center, with methanotrophs showing the opposite pattern (Fig. 5).

**4.2 Spatial heterogeneity of methane emissions**

In dammed rivers, riverbed sediment accumulation creates potential methane emission hotspots. In this study, however, high methane emissions were only observed at the island center, with a ring-like low methane emission zone or even methane sink appearing around the island edge (Fig. 4a). This was attributed to the spatial heterogeneity of methanogens and methanotrophs across the island (Fig. 5), leading to an increase in methane production and a decrease in methane consumption from the island edge to the center. The methane sink at the island edge (Fig. 4) was mainly attributed to the strong oxidation by methanotrophs, which may consume methane to below equilibrium with the atmosphere. Methane emissions may not only rely on bacterial abundance but also bacterial activity (Schwarz et al., 2008). This deserves further studies using other molecular biology techniques, such as DNA/RNA-based stable isotope probing (Dumont and Murrell, 2005). In addition, groundwater DOC and sediment OC at the island edge, which are carbon sources for methane emission, were higher than that at the island center (Fig. 2b,c), suggesting that both sediment heterogeneity and dilution effects of hyporheic exchange had limited contribution to the spatial pattern of methane emissions in the island.

**4.3 Implications**

Greenhouse gas emissions significantly detract from the green credentials of hydropower, and have thus received considerable research attention (Giles, 2006; Hu and Cheng, 2013). Previous studies have revealed that damming causes significant retention of carbon and creates deep, anoxic sediment strata, fueling methanogenesis and net water-air methane flux (Maeck et al., 2013). This study demonstrated that methane emissions at the most area of the sediment-deposited island were generally lower than the adjacent reservoir under water level fluctuation induced by reservoir operation (Fig. 4a), but higher than the drawdown area at other reservoir bank (-0.08–

0.66 mg $h^{-1}m^{-2}$), such as Three Gorge Reservoir (Chen et al., 2011). This was mainly due to the deep sediment strata (about 60 m in depth) in the island. Given the widely distributed sediment-deposited islands in reservoirs, it should be of concern in future estimations of greenhouse gas emissions from dammed rivers.

Until now, few studies have concentrated on organic carbon mineralization in the drawdown area in reservoirs, with most focusing on the process of denitrification (Zarnetske et al., 2011b). Carbon emissions in the drawdown area are poorly understood, especially in regulated and dammed rivers. This study fills the knowledge gap and adds to our understanding of the ecological impacts of hydropower exploitation. Under water level fluctuation induced by reservoir operation, variable oxygen conditions and methane production may also affect the mercury cycle in the drawdown area and thereby the release of methylmercury (a bioaccumulative environmental toxicant) to the river (Marvin-DiPasquale et al., 2009), a subject deserving of further study.

**324 5 Conclusions**

In dammed rivers, sediment deposited islands are widely distributed in sidebays and are potential hotspots of methane emission to the atmosphere. In this study, high methane fluxes were only observed at the island center, while a ring-like zone of low methane emission or even sink was found in the drawdown area around the island edge. We attribute this spatial heterogeneity of methane emissions to hyporheic exchange between the reservoir and island. Under reservoir operation, frequent water level fluctuations drove hyporheic exchange, creating oxygen gradients along the hyporheic flowpath. These oxygen gradients affected the microbial communities associated with methane production and consumption, producing the spatial heterogeneity in methane emissions across sediment-deposited islands. This study will help us to evaluate the global warming effects of hydropower systems.

**Data availability**

The data presented here can be obtained upon request to Wenqing Shi (wqshi@nhri.cn).

**Author contribution**

Qiuwen Chen designed the research; Wenqing Shi, Yuyu Ji and Yuchen Chen performed the research; Cheng Chen and Juhua Yu contributed new reagents/analytic tools; Qiuwen Chen and Wenqing Shi analyzed the data; Jianyun Zhang and Bryce R. Van Dam contributed significant discussions and inputs; Wenqing Shi and Qiuwen Chen wrote the paper with input from all authors.

**Competing interests**

The authors declare that they have no conflict of interest.

**Acknowledgements**

Funding for this study was provided by the National Nature Science Foundation of China (No. 91547206, 51425902, 51709181 and 51709182).

[Figure]

**Fig. 1** Map of the studied island in Manwan reservoir, Lancang-Mekong River. (a) The island site in the river; (b) The monitoring wells, sediment and gas sampling sites on the island.

[Figure]

**Fig. 2** Physicochemical properties of the island and reservoir, including water (a) DO, (b) DOC, and (c) sediment OC. DO = dissolved oxygen, DOC = dissolved oxygen carbon, OC = organic carbon, R = reservoir, W = monitoring wells. Error bars indicate standard deviations.

[Figure]

**Fig. 3** Vertical water level fluctuation (a) and horizontal hyporheic flow rate (b). R =

reservoir, W = monitoring wells. Positive fluxes indicate net flow from the reservoir to island, whereas negative values indicate net flow from the island to reservoir.

[Figure]

**Fig. 4** Methane emissions from the island and reservoir. (a) Spatial pattern of methane emissions; (b) Percentage of the island area emitting methane at a certain flux. Methane fluxes were interpolated separately for the island and reservoir.

[Figure]

**Fig. 5** Abundances of sediment methanogens and methanotrophs in the island. (a) Spatial patterns of methanogen and methanotroph abundances across the island; (b) The ratio of methanogen to methanotroph abundance at each site. W= monitoring wells. Error bars indicate standard deviations.

[Figure]

**Fig. S1** The river impoundment formed islands in the Lancang-Mekong River. There were about 33 sediment-deposited islands, of which 81.8% are located at the convex bank (a), and 18.2% at the concave bank (b).

[Figure]

**Fig. S2** Island image before (a) and after (b) flooding under reservoir operation.

[Figure]

**Fig. S3** Monitoring wells established from the island edge to the center. Sediment samples were collected adjacent to each well.

[Figure]

**Fig. S4** Sampling sites for methane flux analyses in the island and reservoir, including eight sites in the reservoir and seventeen sites on the island.

[Figure]

**Fig. S5** Bifunctional static chambers (a) for methane flux analyses across the sediment-air interface in the island (b) and water-air interface in the reservoir (c). Styrofoam collar was removable.

Table S1 The data about methane fluxes at each sampling site

| Sites | | The concentration in the chamber ($\times10^{-6}$ mol/L) | | | | | Flux ($\times10^{-6}$ mol/h) | Flux ($\times10^{-6}$ mol/h m$^2$) | Flux (mg/h m$^2$) | Average flux (mg/h m$^2$) |
|---|---|---|---|---|---|---|---|---|---|---|
| | | 0 min | 10 min | 20 min | 30 min | 40 min | | | | |
| R1 | Rep-1 | 0.452 | 0.773 | 1.174 | 1.375 | 1.656 | 1.806 | 57.5 | 0.92 | |
| | Rep-2 | 0.641 | 0.921 | 1.280 | 1.680 | 2.160 | 2.279 | 72.6 | 1.16 | 1.07 |
| | Rep-3 | 0.541 | 1.106 | 1.272 | 1.637 | 2.103 | 2.193 | 69.8 | 1.12 | |
| R2 | Rep-1 | 0.797 | 0.887 | 1.232 | 1.778 | 2.081 | 2.075 | 66.1 | 1.06 | |
| | Rep-2 | 1.166 | 1.431 | 1.722 | 2.112 | 2.778 | 2.343 | 74.6 | 1.19 | 1.13 |
| | Rep-3 | 0.823 | 0.869 | 1.197 | 3.825 | 4.273 | NA | NA | NA | |
| R3 | Rep-1 | 0.678 | 1.075 | 1.809 | 2.243 | 3.264 | 3.804 | 121.1 | 1.94 | |
| | Rep-2 | 0.822 | 1.261 | 2.282 | 2.702 | 3.704 | 4.323 | 137.7 | 2.20 | 2.10 |
| | Rep-3 | 1.243 | 1.543 | 2.246 | 2.969 | 4.043 | 4.216 | 134.3 | 2.15 | |
| R4 | Rep-1 | 0.573 | 0.660 | 0.759 | 1.018 | 1.189 | 0.954 | 30.4 | 0.49 | |
| | Rep-2 | 0.390 | 0.553 | 0.665 | 0.797 | 0.829 | 0.673 | 21.4 | 0.34 | 0.39 |
| | Rep-3 | 0.476 | 0.622 | 0.762 | 0.863 | 0.909 | 0.665 | 21.2 | 0.34 | |
| R5 | Rep-1 | 1.867 | 2.871 | 4.802 | 6.432 | 8.739 | 10.383 | 330.7 | 5.29 | |
| | Rep-2 | 1.294 | 2.554 | 4.565 | 6.582 | 9.349 | 12.082 | 384.8 | 6.16 | 5.47 |
| | Rep-3 | 1.188 | 2.165 | 4.289 | 4.913 | 7.935 | 9.745 | 310.4 | 4.97 | |
| R6 | Rep-1 | 0.493 | 1.175 | 1.608 | 2.142 | 2.678 | 3.203 | 102.0 | 1.63 | |
| | Rep-2 | 0.881 | 1.411 | 1.881 | 2.751 | 3.561 | 4.021 | 128.1 | 2.05 | 1.84 |
| | Rep-3 | 0.677 | 1.432 | 1.381 | 2.988 | 3.463 | NA | NA | NA | |
| R7 | Rep-1 | 0.388 | 0.802 | 0.923 | 1.224 | 1.482 | 1.566 | 49.9 | 0.80 | |
| | Rep-2 | 0.653 | 0.876 | 1.875 | 1.896 | 2.165 | NA | NA | NA | 0.84 |
| | Rep-3 | 0.794 | 0.930 | 1.420 | 1.609 | 1.902 | 1.737 | 55.3 | 0.89 | |
| R8 | Rep-1 | 0.971 | 1.626 | 2.515 | 3.401 | 4.015 | 4.718 | 150.3 | 2.40 | |
| | Rep-2 | 0.879 | 0.997 | 1.377 | 1.995 | 2.469 | 2.508 | 79.9 | 1.28 | 1.84 |
| | Rep-3 | 0.726 | 0.726 | 1.543 | 1.945 | 2.807 | 3.605 | 114.8 | 1.84 | |
| 1 | Rep-1 | 0.463 | 0.434 | 0.358 | 0.295 | 0.200 | -0.399 | -12.7 | -0.20 | |
| | Rep-2 | 0.454 | 0.414 | 0.358 | 0.255 | 0.157 | -0.452 | -14.4 | -0.23 | -0.21 |
| | Rep-3 | 0.386 | 0.314 | 0.258 | 0.214 | 0.139 | -0.356 | -11.3 | -0.18 | |
| 2 | Rep-1 | 0.878 | 1.213 | 1.556 | 2.009 | 2.810 | 2.796 | 89.0 | 1.42 | |
| | Rep-2 | 0.784 | 0.970 | 1.432 | 1.945 | 2.442 | 2.575 | 82.0 | 1.31 | 1.26 |
| | Rep-3 | 0.993 | 1.187 | 1.534 | 1.898 | 2.358 | 2.065 | 65.8 | 1.05 | |
| 3 | Rep-1 | 1.068 | 1.483 | 2.425 | 3.567 | 4.738 | 5.654 | 180.1 | 2.88 | |
| | Rep-2 | 1.282 | 1.919 | 2.799 | 3.304 | 4.482 | 4.671 | 148.8 | 2.38 | 2.78 |
| | Rep-3 | 1.482 | 1.949 | 2.876 | 3.967 | 5.519 | 6.055 | 192.8 | 3.09 | |
| 4 | Rep-1 | 2.893 | 6.897 | 8.251 | 12.608 | 16.811 | 20.13 | 641.1 | 10.26 | |
| | Rep-2 | 3.892 | 7.996 | 10.451 | 15.906 | 17.211 | 20.73 | 660.2 | 10.56 | 10.41 |
| | Rep-3 | 1.278 | 2.301 | 4.061 | 5.868 | 8.291 | NA | NA | NA | |
| 5 | Rep-1 | 0.786 | NA | 2.784 | 4.324 | NA | NA | NA | NA | |
| | Rep-2 | 0.876 | 1.805 | 3.180 | 4.625 | 5.769 | 7.564 | 240.9 | 3.85 | 4.27 |
| | Rep-3 | 1.069 | 2.075 | 3.208 | 5.141 | 7.203 | 9.201 | 293.0 | 4.69 | |
| 6 | Rep-1 | 1.221 | 1.519 | 1.697 | 1.875 | 1.933 | 1.068 | 34.0 | 0.54 | 0.49 |

| | | | | | | | | | |
|---|---|---|---|---|---|---|---|---|---|
| | Rep-2 | 0.782 | 0.819 | 0.999 | 1.153 | 1.338 | 0.868 | 27.6 | 0.44 | |
| | Rep-3 | 1.887 | 1.267 | 1.756 | 2.154 | 2.375 | NA | NA | NA | |
| 7 | Rep-1 | 0.456 | 0.476 | 0.481 | 0.493 | 0.507 | 0.071 | 2.3 | 0.04 | 0.03 |
| | Rep-2 | 0.479 | 0.489 | 0.499 | 0.505 | 0.515 | 0.05 | 1.6 | 0.03 | |
| | Rep-3 | 0.578 | 0.587 | 0.602 | 0.611 | 0.62 | 0.065 | 2.1 | 0.03 | |
| 8 | Rep-1 | 0.399 | 0.408 | 0.417 | 0.422 | 0.428 | 0.043 | 1.4 | 0.02 | 0.03 |
| | Rep-2 | 0.456 | 0.471 | 0.482 | 0.503 | 0.512 | 0.086 | 2.7 | 0.04 | |
| | Rep-3 | 0.488 | 0.497 | 0.505 | 0.517 | 0.519 | 0.049 | 1.6 | 0.02 | |
| 9 | Rep-1 | 0.678 | 0.6768 | 0.676 | 0.675 | 0.6745 | -0.005 | -0.2 | 0.00 | -0.01 |
| | Rep-2 | 0.544 | 0.536 | 0.531 | 0.528 | 0.521 | -0.032 | -1.0 | -0.02 | |
| | Rep-3 | 0.487 | 0.483 | 0.48 | 0.475 | 0.472 | -0.023 | -0.7 | -0.01 | |
| 10 | Rep-1 | 0.661 | 0.67 | 0.678 | 0.683 | 0.687 | 0.039 | 1.2 | 0.02 | 0.02 |
| | Rep-2 | 0.495 | 0.502 | 0.506 | 0.512 | 0.519 | 0.035 | 1.1 | 0.02 | |
| | Rep-3 | 0.785 | 0.791 | 0.795 | 0.802 | 0.809 | 0.0355 | 1.1 | 0.02 | |
| 11 | Rep-1 | 0.555 | 0.5545 | 0.5538 | 0.5535 | 0.5528 | -0.003 | -0.1 | 0.00 | 0.00 |
| | Rep-2 | 0.453 | 0.454 | 0.454 | 0.455 | 0.456 | 0.004 | 0.1 | 0.00 | |
| | Rep-3 | 0.768 | 0.745 | 0.755 | 0.777 | 0.815 | NA | NA | NA | |
| 12 | Rep-1 | 0.787 | 1.199 | 1.443 | 2.108 | 2.588 | 2.707 | 86.2 | 1.38 | 1.12 |
| | Rep-2 | 0.722 | 0.805 | 1.119 | 1.403 | 1.844 | 1.705 | 54.3 | 0.87 | |
| | Rep-3 | 0.477 | 0.674 | 0.846 | 0.876 | 1.532 | NA | NA | NA | |
| 13 | Rep-1 | 0.491 | 1.118 | 1.694 | 2.171 | 2.847 | 3.46 | 110.2 | 1.76 | 1.61 |
| | Rep-2 | 0.493 | 1.015 | 1.290 | 1.965 | 2.390 | 2.85 | 90.8 | 1.45 | |
| | Rep-3 | NA | 0.564 | NA | 0.762 | 1.438 | NA | NA | NA | |
| 14 | Rep-1 | 0.393 | 0.444 | 0.477 | 0.521 | 0.532 | 0.021 | 0.7 | 0.01 | 0.01 |
| | Rep-2 | 0.420 | 0.425 | 0.430 | NA | 0.439 | 0.029 | 0.9 | 0.01 | |
| | Rep-3 | 0.422 | 0.427 | 0.432 | 0.435 | 0.440 | 0.0265 | 0.8 | 0.01 | |
| 15 | Rep-1 | 0.754 | 0.733 | 0.695 | 0.677 | 0.644 | -0.166 | -5.3 | -0.08 | -0.05 |
| | Rep-2 | 0.472 | 0.463 | 0.455 | 0.447 | 0.435 | -0.054 | -1.7 | -0.03 | |
| | Rep-3 | 0.511 | 0.493 | 0.475 | 0.457 | 0.447 | -0.098 | -3.1 | -0.05 | |
| 16 | Rep-1 | 0.567 | 0.540 | 0.525 | 0.511 | 0.503 | -0.094 | -3.0 | -0.05 | -0.04 |
| | Rep-2 | 0.853 | 0.840 | 0.835 | 0.821 | 0.809 | -0.064 | -2.0 | -0.03 | |
| | Rep-3 | 0.454 | 0.440 | 0.435 | 0.411 | 0.398 | -0.085 | -2.7 | -0.04 | |
| 17 | Rep-1 | 0.263 | 0.564 | 0.672 | 0.675 | 0.785 | NA | NA | NA | 0.01 |
| | Rep-2 | 0.567 | 0.570 | 0.572 | 0.574 | 0.575 | 0.012 | 0.4 | 0.01 | |
| | Rep-3 | 0.368 | 0.370 | 0.375 | 0.377 | 0.381 | 0.02 | 0.6 | 0.01 | |